# Maximum-Likelihood Inverse Reinforcement Learning with Finite-Time Guarantees

**Siliang Zeng**
University of Minnesota, Twin Cities
Minneapolis, MN, USA
`zeng0176@umn.edu`

**Chenliang Li**
The Chinese University of Hong Kong,
Shenzhen, China
`chenliangli@link.cuhk.edu.cn`

**Alfredo Garcia**
Texas A&M University
College Station, TX, USA
`alfredo.garcia@tamu.edu`

**Mingyi Hong**
University of Minnesota, Twin Cities
Minneapolis, MN, USA
`mhong@umn.edu`

## Abstract

Inverse reinforcement learning (IRL) aims to recover the reward function and the associated optimal policy that best fits observed sequences of states and actions implemented by an expert. Many algorithms for IRL have an inherently nested structure: the inner loop finds the optimal policy given parametrized rewards while the outer loop updates the estimates towards optimizing a measure of fit. For high dimensional environments such nested-loop structure entails a significant computational burden. To reduce the computational burden of a nested loop, novel methods such as SQIL [1] and IQ-Learn [2] emphasize policy estimation at the expense of reward estimation accuracy. However, without accurate estimated rewards, it is not possible to do counterfactual analysis such as predicting the optimal policy under different environment dynamics and/or learning new tasks. In this paper we develop a novel *single-loop* algorithm for IRL that does not compromise reward estimation accuracy. In the proposed algorithm, each policy improvement step is followed by a stochastic gradient step for likelihood maximization. We show that the proposed algorithm provably converges to a stationary solution with a finite-time guarantee. If the reward is parameterized linearly, we show the identified solution corresponds to the solution of the maximum entropy IRL problem. Finally, by using robotics control problems in MuJoCo and their transfer settings, we show that the proposed algorithm achieves superior performance compared with other IRL and imitation learning benchmarks.

## 1 Introduction

Given observed trajectories of states and actions implemented by an expert, we consider the problem of estimating the reinforcement learning environment in which the expert was trained. This problem is generally referred to as inverse reinforcement learning (IRL) (see [3] for a recent survey). Assuming the environment dynamics are known (or available online), the IRL problem consists of estimating the reward function and the expert's policy (optimizing such rewards) that best fits the data. While there are limitations on the identifiability of rewards [4], the estimation of rewards based upon expert trajectories enables important counterfactual analysis such as the estimation of optimal policies under *different* environment dynamics and/or reinforcement learning of *new* tasks.

In the seminal work [5], the authors developed an IRL formulation, in which the model for the expert's behavior is the policy that maximizes entropy subject to a constraint requiring that the expected

36th Conference on Neural Information Processing Systems (NeurIPS 2022).

features under such policy match the empirical averages in the expert's observation dataset. The algorithms developed for MaxEnt-IRL [5–7] have a nested loop structure, alternating between an outer loop with a reward update step, and an inner loop that calculates the explicit policy estimates. The computational burden of this nested structure is manageable in tabular environments, but it becomes significant in high dimensional settings requiring function approximation.

Towards developing more efficient IRL algorithms, a number of works [8–12] propose to leverage the idea of adversarial training [13]. These algorithms learn a non-stationary reward function through training a discriminator, which is then used to guide the policy to match the behavior trajectories from the expert dataset. However, [14] pointed out that the resulting discriminator (hence the reward function) typically cannot be used in new learning tasks, since it is highly dependent on the corresponding policy and current environment dynamics. Moreover, due to the brittle approximation techniques and sensitive hyperparameter choice in the adversarial training, these IRL algorithms can be unstable. [15, 16].

More recent works [1, 2] have developed algorithms to alleviate the computational burden of the nested-loop training procedures. In [1], the authors propose to model the IRL using certain maximum entropy RL problem with specific reward function (which assigns $r = +1$ for matching expert demonstrations and $r = 0$ for all other behaviors). Then a soft Q imitation learning (SQIL) algorithm is developed. In [2], the authors propose to transform the standard formulation of IRL (discussed above) into a single-level problem, through learning a soft Q-function to implicitly represent the reward function and the policy. An inverse soft-Q learning (IQ-Learn) algorithm is then developed, which is shown to be effective in estimating the policy for the environment that it is trained on. Despite being computationally efficient, IQ-Learn sacrifices the accuracy in estimating the rewards since it indirectly recovers rewards from a soft Q-function approximator which is highly dependent upon the environment dynamics and does not strictly satisfy the soft-Bellman equation. Therefore it is not well-suited for counterfactual prediction or transfer learning setting.

Finally, in $f$-IRL [14] the authors consider an approach for estimating rewards based on the minimization of several measures of divergence with respect to the expert's state visitation measure. The approach is limited to estimating rewards that only depend on state. Moreover, while the results reported are based upon a single-loop implementation, the paper does not provide a convergence guarantee to support performance. We refer the readers to Appendix A for other related works.

**Our Contributions.** The goal of this work is to develop an algorithm for IRL which is capable of producing high-quality estimates of *both* rewards and behavior policies with finite-time guarantees. The major contributions of this work are listed below.

• We consider a formulation of IRL based on maximum likelihood (ML) estimation over optimal (entropy-regularized) policies, and prove that a strong duality relationship with maximum entropy IRL holds if rewards are represented by a *linear* combination of features. [1] The ML formulation is a *bi-level* optimization problem, where the upper-level problem maximizes the likelihood function, while the lower-level finds the optimal policy under the current reward parameterization. Such a bi-level structure is not only instrumental to the subsequent algorithm design, but is also flexible to incorporate the use of state-only, as well as the regular reward function (which depends on the state and action pair). The former is suitable for transfer learning since it is *insensitive* to the changes of the environment dynamics, while the latter can be used to efficiently imitate the expert policy.

• Based on the ML-IRL formulation, we develop an efficient algorithm. To avoid the computational burden of repeatedly solving the lower-level policy optimization problem, the proposed algorithm has a single-loop structure where the policy improvement step and reward optimization step are performed alternatingly so that each step can be performed relatively cheaply. Further, we show that the algorithm has strong theoretical guarantees: to achieve certain $\epsilon$-approximate stationary solution for a non-linearly parameterized problem, it requires $\mathcal{O}(\epsilon^{-2})$ steps of policy and reward updates each. To our knowledge, it is the first algorithm which has finite-time guarantee for the IRL problem under nonlinear parameterization of reward functions.

• We conduct extensive experiments to demonstrate that the proposed algorithm outperforms many state-of-the-art IRL algorithms in *both* policy estimation and reward recovery. In particular, when

---

[1]Heuristic arguments for this duality result are discussed in [5] wherein the distribution of state-action paths is approximated (see equation (4) in [5]) and the equivalence between maximum entropy estimation and maximum likelihood (over the class of exponential distributions) [17] is invoked.

transferring to a new environment, RL algorithms using rewards recovered by the proposed algorithm outperform those that use rewards recovered from existing IRL and imitation learning benchmarks.

## 2 Preliminaries

In this section, we review the fundamentals of the maximum entropy inverse reinforcement learning (MaxEnt-IRL). We consider an MDP defined by the tuple $(\mathcal{S}, \mathcal{A}, \mathcal{P}, \eta, r, \gamma)$; $\mathcal{S}$ and $\mathcal{A}$ denote the state space and the action space respectively; $\mathcal{P}(s'|s, a) : \mathcal{S} \times \mathcal{A} \times \mathcal{S} \to [0, 1]$ denotes the transition probability; $\eta(\cdot)$ denotes the distribution for the initial state; $r(s, a) : \mathcal{S} \times \mathcal{A} \to R$ is the reward function and $\gamma$ is a discount factor.

The MaxEnt-IRL formulation [6, 18–20] consists of finding a policy maximizing entropy subject to the expected features under such policy matching the empirical averages in the expert's observation dataset. Specifically, the MaxEnt-IRL formulation is given by:

$$\max_{\pi} \quad H(\pi) := \mathbb{E}_{\tau \sim \pi}\left[\sum_{t=0}^{\infty} -\gamma^t \log \pi(a_t|s_t)\right] \qquad \text{(MaxEnt-IRL)}$$

$$\text{s.t.} \quad \mathbb{E}_{\tau \sim \pi}\left[\sum_{t=0}^{\infty} \gamma^t \phi(s_t, a_t)\right] = \mathbb{E}_{\tau \sim \pi^{\mathrm{E}}}\left[\sum_{t=0}^{\infty} \gamma^t \phi(s_t, a_t)\right]$$

where $\tau = \{(s_t, a_t)\}_{t=0}^{\infty}$ denotes a trajectory, $\phi(s, a)$ is the feature vector of the state-action pair $(s, a)$ and $\pi^{\mathrm{E}}$ denotes the expert policy. Let $\theta$ denote the dual variable for the linear constraint, then the Lagrangian of (MaxEnt-IRL) is given by

$$\mathcal{L}(\pi, \theta) := H(\pi) + \left\langle \theta, \mathbb{E}_{\tau \sim \pi}\left[\sum_{t=0}^{\infty} \gamma^t \phi(s_t, a_t)\right] - \mathbb{E}_{\tau \sim \pi^{\mathrm{E}}}\left[\sum_{t=0}^{\infty} \gamma^t \phi(s_t, a_t)\right] \right\rangle. \qquad (1)$$

In [6, 18, 19], the authors proposed a "dual descent" algorithm, which alternates between *i)* solving $\max_{\pi} \mathcal{L}(\pi, \theta)$ for fixed $\theta$, and *ii)* a gradient descent step to optimize the dual variable $\theta$. It is shown that the optimizer $\pi_{\theta}^*$ in step *i)* can be recursively defined as $\pi_{\theta}^*(a_t|s_t) = \frac{Z_{a_t|s_t,\theta}}{Z_{s_t,\theta}}$, where $\log Z_{a_t|s_t,\theta} = \phi(s_t, a_t)^T \theta + \gamma \mathbb{E}_{s_{t+1} \sim \mathcal{P}(\cdot|s_t, a_t)}\left[\log Z_{s_{t+1},\theta}\right]$ and $\log Z_{s_t,\theta} = \log\left(\sum_{a \in \mathcal{A}} Z_{a|s_t,\theta}\right)$.

From a computational perspective, the above algorithm is not efficient: it has a nested-loop structure, which repeatedly computes the optimal policy $\pi_{\theta}^*$ under each variable $\theta$. It is known that when the underlying MDP is of high-dimension, such an algorithm can be computationally prohibitive [9, 10].

Recent work [2] proposed an algorithm called IQ-Learn to improve upon the MaxEnt-IRL by considering a saddle-point formulation:

$$\min_r \max_{\pi} \left\{ H(\pi) + \mathbb{E}_{\tau \sim \pi}\left[\sum_{t=0}^{\infty} \gamma^t \cdot r(s_t, a_t)\right] - \mathbb{E}_{\tau \sim \pi^{\mathrm{E}}}\left[\sum_{t=0}^{\infty} \gamma^t \cdot r(s_t, a_t)\right] \right\} \qquad (2)$$

where $r(s_t, a_t)$ is the reward associated with state-action pair $(s_t, a_t)$. The authors show that this problem can be transformed into an optimization problem *only* defined in terms of the soft Q-function, which implicitly represents both reward and policy. IQ-Learn is shown to be effective in imitating the expert behavior while only relying on the estimation of the soft Q-function. However, the implicit reward estimate obtained is not necessarily accurate since its soft Q-function estimate depends on the environment dynamics and does not strictly satisfy the soft-Bellman equation. Hence, it is difficult to transfer the recovered reward function to new environments.

## 3 Problem Formulation

In this section, we consider a ML formulation of the IRL problem and formalize a duality relationship with maximum entropy-based formulation (MaxEnt-IRL).

### Maximum Log-Likelihood IRL (ML-IRL)

A model of the expert's behavior is a randomized policy $\pi_{\theta}(\cdot|s)$, where $\pi_{\theta}$ is a specific policy corresponding to the reward parameter $\theta$. With the state dynamics $\mathcal{P}(s_{t+1}|s_t, a_t)$, the discounted

log-likelihood of observing the expert trajectory $\tau$ under model $\pi_\theta$ can be written follows:

$$\mathbb{E}_{\tau \sim \pi^{\mathrm{E}}}\Big[\log \prod_{t \geq 0}(\mathcal{P}(s_{t+1}|s_t, a_t)\pi_\theta(a_t|s_t))^{\gamma^t}\Big] = \mathbb{E}_{\tau \sim \pi^{\mathrm{E}}}\Big[\sum_{t \geq 0} \gamma^t \log \pi_\theta(a_t|s_t)\Big]$$

$$+ \mathbb{E}_{\tau \sim \pi^{\mathrm{E}}}\Big[\sum_{t \geq 0} \gamma^t \log \mathcal{P}(s_{t+1}|s_t, a_t)\Big].$$

Then we consider the following maximum log-likelihood IRL formulation:

$$\max_\theta \quad L(\theta) := \mathbb{E}_{\tau \sim \pi^{\mathrm{E}}}\Big[\sum_{t=0}^\infty \gamma^t \log \pi_\theta(a_t|s_t)\Big] \tag{ML-IRL}$$

$$s.t \quad \pi_\theta := \arg\max_\pi \; \mathbb{E}_{\tau \sim \pi}\bigg[\sum_{t=0}^\infty \gamma^t \Big(r(s_t, a_t; \theta) + \mathcal{H}(\pi(\cdot|s_t))\Big)\bigg], \tag{3a}$$

where $r(s, a; \theta)$ is the reward function and $\mathcal{H}(\pi(\cdot|s)) := -\sum_{a \in \mathcal{A}} \pi(a|s) \log \pi(a|s)$.

We now make some remarks about ML-IRL. First, the problem takes the form of a *bi-level* optimization problem, where the *upper-level* problem (ML-IRL) optimizes the reward parameter $\theta$, while the *lower-level* problem describes the corresponding policy $\pi_\theta$ as the solution to an entropy-regularized MDP ([21, 22]). In what follows we will leverage recently developed (stochastic) algorithms for bi-level optimization [23–25], that avoid the high complexity resulted from nested loop algorithms. Second, it is reasonable to use the ML function as the loss, because it searches for a reward function which generates a behavior policy that can best fit the expert demonstrations. While the ML function has been considered in [26, 27], they rely on heuristic algorithms with nested-loop computations to solve their IRL formulations, and the theoretical properties are not studied. Finally, the lower-level problem has been well-studied in the literature [21, 22, 28–30]. The entropy regularization in (3a) ensures the uniqueness of the optimal policy $\pi_\theta$ under the fixed reward function $r(s, a; \theta)$ [21, 28]. Even when the underlying MDP is high-dimensional and/or complex, the optimal policy could still be obtained; see recent developments in [21, 22]. We close this section by formally establishing a connection between (MaxEnt-IRL) and (ML-IRL).

**Theorem 1. (Strong Duality)** *Suppose that the reward function is given as: $r(s, a; \theta) := \phi(s, a)^T \theta$, for all $s \in \mathcal{S}$ and $a \in \mathcal{A}$. Then (ML-IRL) is the Lagrangian dual of (MaxEnt-IRL). Furthermore, strong duality holds, that is: $L(\theta^*) = H(\pi^*)$, where $\theta^*$ and $\pi^*$ are the global optimal solutions for problems (ML-IRL) and (MaxEnt-IRL), respectively.*

The proof of Theorem 1 is relegated to Appendix G. To our knowledge this result which specifically addresses the (MaxEnt-IRL) formulation is novel. Under finite horizon, a duality between ML estimation and maximum causal entropy is obtained in [18, Theorem 3]. However, the problem considered in that paper is not in RL nor IRL setting, therefore they cannot be directly used in the context of the present paper.

The above duality result reveals a strong connection between the two formulations under linear reward parameterization. Due to the duality result, we know that (ML-IRL) is a concave problem under linear reward parameterization. In this case, any stationary solution to (ML-IRL) is a global optimal estimator of the reward parameter.

## 4 The Proposed Algorithm

In this section, we design algorithms for (ML-IRL). Recall that one major drawback of algorithms for (MaxEnt-IRL) is that, they repeatedly solve certain policy optimization problem in the inner loop. Even though the recently proposed algorithm IQ-Learn [2] tries to improve the computational efficiency through implicitly representing the reward function and the policy by a Q-function approximator, it has sacrificed the estimation accuracy of the recovered reward. Therefore, one important goal of our design is to find provably efficient algorithms that can avoid high-complexity operations and accurately recover the reward function. Specifically, it is desirable that the resulting algorithm only uses a finite number of reward and policy updates to reach certain high-quality solutions.

To proceed, we will leverage the special *bi-level* structure of the ML-IRL problem. The idea is to alternate between one step of policy update to improve the solution of the lower-level problem, and

---

**Algorithm 1** *Maximum Likelihood Inverse Reinforcement Learning (ML-IRL)*

---

**Input:** Initialize reward parameter $\theta_0$ and policy $\pi_0$. Set the reward parameter's stepsize as $\alpha$.

**for** $k = 0, 1, \ldots, K - 1$ **do**

    **Policy Evaluation:** Compute $Q_{r_{\theta_k}, \pi_k}^{\text{soft}}(\cdot, \cdot)$ under reward function $r(\cdot, \cdot; \theta_k)$

    **Policy Improvement:** $\pi_{k+1}(\cdot|s) \propto \exp(Q_{r_{\theta_k}, \pi_k}^{\text{soft}}(s, \cdot)), \forall s \in \mathcal{S}$.

    **Data Sampling I:** Sampling an expert trajectory $\tau_k^{\text{E}} := \{s_t, a_t\}_{t \geq 0}$

    **Data Sampling II:** Sampling a trajectory $\tau_k^{\text{A}} := \{s_t, a_t\}_{t \geq 0}$ from the current policy $\pi_{k+1}$

    **Estimating Gradient:** $g_k := h(\theta_k; \tau_k^{\text{E}}) - h(\theta_k; \tau_k^{\text{A}})$ where $h(\theta; \tau) := \sum_{t \geq 0} \gamma^t \nabla_\theta r(s_t, a_t; \theta)$

    **Reward Parameter Update:** $\theta_{k+1} := \theta_k + \alpha g_k$

**end for**

---

one step of the parameter update which improves the upper-level loss function. At each iteration $k$, given the current policy $\pi_k$ and the reward parameter $\theta_k$, a new policy $\pi_{k+1}$ is generated from the policy improvement step, and $\theta_{k+1}$ is generated by the reward optimization step.

This kind of alternating update is efficient, because there is no need to completely solve the policy optimization subproblem, before updating the reward parameters. It has been used in many other RL related settings as well. For example, the well-known actor-critic (AC) algorithm for policy optimization [31, 32, 23] alternates between one step of policy update, and one step of critic parameter update. Below we present the details of our algorithm at a given iteration $k$.

**Policy Improvement Step.** Let us consider optimizing the lower-level problem, when the reward parameter $\theta_k$ is held fixed. Towards this end, define the so-called soft Q and soft value functions for a given policy $\pi_k$ and a reward parameter $\theta_k$:

$$V_{r_k, \pi_k}^{\text{soft}}(s) = \mathbb{E}_{\pi_k} \left[ \sum_{t=0}^{\infty} \gamma^t \left( r(s_t, a_t; \theta_k) + \mathcal{H}(\pi_k(\cdot|s_t)) \right) \middle| s_0 = s \right] \tag{4a}$$

$$Q_{r_k, \pi_k}^{\text{soft}}(s, a) = r(s, a; \theta_k) + \gamma \mathbb{E}_{s' \sim \mathcal{P}(\cdot|s, a)} \left[ V_{r_k, \pi_k}^{\text{soft}}(s) \right] \tag{4b}$$

We will adopt the well-known *soft policy iteration* [21] to optimize the lower-level problem (3a). Under the current reward parameter $\theta_k$ and the policy $\pi_k$, the soft policy iteration generates a new policy $\pi_{k+1}$ as follows

$$\pi_{k+1}(a|s) \propto \exp\left( Q_{r_{\theta_k}, \pi_k}^{\text{soft}}(s, a) \right), \quad \forall s \in \mathcal{S}, a \in \mathcal{A}. \tag{5}$$

Under a fixed reward function, it can be shown that the new policy $\pi_{k+1}$ monotonically improves $\pi_k$, and it converges linearly to the optimal policy; see [21, Theorem 4] and [28, Thoerem 1].

Note that in practice, we usually do not have direct access to the exact soft Q-function in (4b). In order to perform the policy improvement, a few stochastic update steps in soft Q-learning [21] or soft Actor-Critic (SAC) [22] could be used to replace the one-step soft policy iteration (5). In the appendix, we present Alg. 2 to demonstrate such practical implementation of our proposed algorithm.

**Reward Optimization Step.** We propose to use a stochastic gradient-type algorithm to optimize $\theta$. Towards this end, let us first derive the exact gradient $\nabla L(\theta)$. See Appendix D for detailed proof.

**Lemma 1.** *The gradient of the likelihood function $L(\theta)$ can be expressed as follows:*

$$\nabla L(\theta) = \mathbb{E}_{\tau \sim \pi^{\text{E}}} \left[ \sum_{t \geq 0} \gamma^t \nabla_\theta r(s_t, a_t; \theta) \right] - \mathbb{E}_{\tau \sim \pi_\theta} \left[ \sum_{t \geq 0} \gamma^t \nabla_\theta r(s_t, a_t; \theta) \right]. \tag{6}$$

To obtain stochastic estimators of the exact gradient $\nabla L(\theta_k)$, we take two approximation steps: 1) approximate the optimal policy $\pi_{\theta_k}$ by $\pi_{k+1}$ in (5), since the optimal policy $\pi_{\theta_k}$ is not available throughout the algorithm; 2) sample one expert trajectory $\tau_k^{\text{E}}$ which is already generated by the expert policy $\pi^{\text{E}}$; 3) sample one trajectory $\tau_k^{\text{A}}$ from the current policy $\pi_{k+1}$.

Following the approximation steps mentioned above, we construct a stochastic estimator $g_k$ to approximate the exact gradient $\nabla L(\theta_k)$ in (6) as follows:

$$g_k := h(\theta_k; \tau_k^{\text{E}}) - h(\theta_k; \tau_k^{\text{A}}), \quad \text{where} \quad h(\theta; \tau) := \sum_{t \geq 0} \gamma^t \nabla_\theta r(s_t, a_t; \theta). \tag{7}$$

With the stochastic gradient estimator $g_k$, the reward parameter $\theta_k$ is updated as:

$$\theta_{k+1} = \theta_k + \alpha g_k. \tag{8}$$

where $\alpha$ is the stepsize in updating the reward parameter.

In summary, the proposed algorithm for solving the ML-IRL problem (ML-IRL) is given in Alg. 1.

## 5 Theoretical Analysis

In this section, we present finite-time guarantees for the proposed algorithm.

To begin with, first recall that in Sec. 3, we have mentioned that (ML-IRL) is a bi-level problem, where the upper level (resp. the lower level) problem optimizes the reward parameter (resp. the policy). In order to solve (ML-IRL), our algorithm 1 has a *singe-loop* structure, which alternates between one step of policy update and one step of the reward parameter update. Such a single-loop structure indeed has computational benefit, but it also leads to potential unstableness, since the lower level problem can stay far away from its true solutions. Specifically, at each iteration $k$, the potential unstableness is induced by the distribution mismatch between the policy $\pi_{k+1}$ and $\pi_{\theta_k}$, when we use estimator $g_k$ (7) to approximate the exact gradient $\nabla L(\theta_k)$ (6) in updating the reward parameter $\theta_k$.

Towards stabilizing the algorithm, we adopt the so-called *two-timescale* stochastic approximation (TTSA) approach [33, 23], where the lower-level problem updates in a faster time-scale (i.e., converges faster) compared with its upper-level counterpart. Intuitively, the TTSA enables the $\pi_{k+1}$ tracks the optimal $\pi_{\theta_k}$, leading to a stable algorithm. In the proposed Algorithm 1, the policy (lower-level variable) is continuously updated by the soft policy iteration (5), and it is 'fast' because it converges linearly to the optimal policy under a fixed reward function [28, Theorem 1]. On the other hand, the reward parameter update (8) does not have such linear convergence property, therefore it works in a 'slow' timescale. To begin our analysis, let us first present a few technical assumptions.

**Assumption 1** (Ergodicity). *For any policy $\pi$, assume the Markov chain with transition kernel $\mathcal{P}$ is irreducible and aperiodic under policy $\pi$. Then there exist constants $\kappa > 0$ and $\rho \in (0,1)$ such that*

$$\sup_{s \in \mathcal{S}} \|\mathbb{P}(s_t \in \cdot | s_0 = s, \pi) - \mu_\pi(\cdot)\|_{TV} \leq \kappa \rho^t, \quad \forall\, t \geq 0$$

*where $\|\cdot\|_{TV}$ is the total variation (TV) norm; $\mu_\pi$ is the stationary state distribution under $\pi$.*

Assumption 1 assumes the Markov chain mixes at a geometric rate. It is a common assumption in the iterature of RL [34, 35, 32], which holds for any time-homogeneous Markov chain with finite-state space or any uniformly ergodic Markov chain with general state space.

**Assumption 2.** *For any $s \in \mathcal{S}$, $a \in \mathcal{A}$ and any reward parameter $\theta$, the following holds:*

$$\|\nabla_\theta r(s, a; \theta)\| \leq L_r, \tag{9a}$$

$$\|\nabla_\theta r(s, a; \theta_1) - \nabla_\theta r(s, a; \theta_2)\| \leq L_g \|\theta_1 - \theta_2\| \tag{9b}$$

*where $L_r$ and $L_g$ are positive constants.*

Assumption 2 assumes that the parameterized reward function has bounded gradient and is Lipschitz smooth. Such assumption in Lipschitz property are common in the literature of min-max / bi-level optimization [36, 23, 37, 25, 38].

Based on Assumptions 1 - 2, we next provide the following Lipschitz properties:

**Lemma 2.** *Suppose Assumptions 1 - 2 hold. For any reward parameter $\theta_1$ and $\theta_2$, the following results hold:*

$$|Q^{soft}_{r_{\theta_1}, \pi_{\theta_1}}(s, a) - Q^{soft}_{r_{\theta_2}, \pi_{\theta_2}}(s, a)| \leq L_q \|\theta_1 - \theta_2\|, \quad \forall s \in \mathcal{S}, a \in \mathcal{A} \tag{10a}$$

$$\|\nabla L(\theta_1) - \nabla L(\theta_2)\| \leq L_c \|\theta_1 - \theta_2\| \tag{10b}$$

*where $Q^{soft}_{r_\theta, \pi_\theta}(\cdot, \cdot)$ denotes the soft Q-function under the reward function $r(\cdot, \cdot; \theta)$ and the policy $\pi_\theta$. The positive constants $L_q$ and $L_c$ are defined in Appendix E.*

The Lipschitz properties identified in Lemma 2 are vital for the convergence analysis. Then we present the main results, which show the convergence speed of the policy $\{\pi_k\}_{k \geq 0}$ and the reward parameter $\{\theta_k\}_{k \geq 0}$ in the Alg. 1. Please see Appendix E for the detailed proof.

**Theorem 2.** *Suppose Assumptions [1] - [2] hold. Selecting stepsize $\alpha := \frac{\alpha_0}{K^\sigma}$ for the reward update step [(8)] where $\alpha_0 > 0$ and $\sigma \in (0,1)$ are some fixed constants, and $K$ is the total number of iterations to be run by the algorithm. Then the following result holds:*

$$\frac{1}{K}\sum_{k=0}^{K-1}\mathbb{E}\left[\left\|\log\pi_{k+1}-\log\pi_{\theta_k}\right\|_\infty\right]=\mathcal{O}(K^{-1})+\mathcal{O}(K^{-\sigma}) \tag{11a}$$

$$\frac{1}{K}\sum_{k=0}^{K-1}\mathbb{E}\left[\|\nabla L(\theta_k)\|^2\right]=\mathcal{O}(K^{-\sigma})+\mathcal{O}(K^{-1+\sigma})+\mathcal{O}(K^{-1}) \tag{11b}$$

*where we denote $\|\log\pi_{k+1}-\log\pi_{\theta_k}\|_\infty := \max_{s\in\mathcal{S},a\in\mathcal{A}}\left|\log\pi_{k+1}(a|s)-\log\pi_{\theta_k}(a|s)\right|$. In particular, setting $\sigma = 1/2$, then both quantities in [(11a)] and [(11b)] converge with the rate $\mathcal{O}(K^{-1/2})$.*

In Theorem [2], we present the finite-time guarantee for the convergence of the Alg.[1]. Moreover, as a special case, when the reward is parameterized as a linear function, we know that ([ML-IRL]) is concave and Theorem [2] provides a stronger guarantee which identify the global optimal reward estimator in finite time.

We provide a proof sketch below to present the key steps. The detailed proof is in Appendix [H].

*Proof sketch.* We outline our main steps in analyzing [(11a)] and [(11b)] respectively.

In order to show the convergence of policy estimates in [(11a)], there are several key steps. First, we note that both policies $\pi_{k+1}$ and $\pi_{\theta_k}$ are in the softmax parameterization, where $\pi_{k+1}(\cdot|s) \propto \exp\left(Q^{\mathrm{soft}}_{r_{\theta_k},\pi_k}(s,\cdot)\right)$ and $\pi_{\theta_k}(\cdot|s) \propto \exp\left(Q^{\mathrm{soft}}_{r_{\theta_k},\pi_{\theta_k}}(s,\cdot)\right)$. Then, we can show a Lipschitz continuity property between the policy and the soft Q-function:

$$\|\log\pi_{k+1}-\log\pi_{\theta_k}\|_\infty \leq 2\|Q^{\mathrm{soft}}_{r_{\theta_k},\pi_k}-Q^{\mathrm{soft}}_{r_{\theta_k},\pi_{\theta_k}}\|_\infty,$$

where the infinity norm $\|\cdot\|_\infty$ is defined over the state-action space $\mathcal{S}\times\mathcal{A}$. Moreover, by analyzing the contraction property of the soft policy iteration [(5)], we bound $\|Q^{\mathrm{soft}}_{r_{\theta_k},\pi_k}-Q^{\mathrm{soft}}_{r_{\theta_k},\pi_{\theta_k}}\|_\infty$ as:

$$\|Q^{\mathrm{soft}}_{r_{\theta_k},\pi_k}-Q^{\mathrm{soft}}_{r_{\theta_k},\pi_{\theta_k}}\|_\infty \leq \gamma\|Q^{\mathrm{soft}}_{r_{\theta_{k-1}},\pi_{k-1}}-Q^{\mathrm{soft}}_{r_{\theta_{k-1}},\pi_{\theta_{k-1}}}\|_\infty + 2L_q\|\theta_k-\theta_{k-1}\|.$$

To ensure that the error term $\|\theta_k - \theta_{k-1}\|$ is small, we select the stepsize of reward parameters as $\alpha := \frac{\alpha_0}{K^\sigma}$, where $K$ is the total number of iterations and $\sigma > 0$. Then, by combining previous two steps, we could further show the convergence rate of the policy estimates in [(11a)].

To prove the convergence of the reward parameters in [(11b)], we first leverage the Lipschitz smooth property of $L(\theta)$ in [(10b)]. However, one technical challenge in the convergence analysis is how to handle the bias between the gradient estimator $g_k$ defined in [(7)] and the exact gradient $\nabla L(\theta_k)$. When we construct the gradient estimator $g_k$ in [(7)], we need to sample trajectories from the current policy $\pi_{k+1}$ and the expert dataset $\mathcal{D}$. However, according to the expression of $\nabla L(\theta_k)$ in [(6)], the trajectories are sampled from the optimal policy $\pi_{\theta_k}$ and the dataset $\mathcal{D}$. Hence, there is a distribution mismatch between $\pi_{k+1}$ and $\pi_{\theta_k}$. Our key idea is to leverage [(11a)] to handle this distribution mismatch error, and thus show that the bias between $g_k$ and $\nabla L(\theta_k)$ could be controlled. $\square$

To the best of our knowledge, Theorem [2] is the first non-asymptotic convergence result for IRL with nonlinear reward parameterization.

## 6   A Discussion over State-Only Reward

In this section we consider the IRL problems modeled by using rewards that are only a function of the state. A lower dimensional representation of the agent's preferences (i.e. in terms only of states as opposed to states *and* actions) is more likely to facilitate counterfactual analysis such as predicting the optimal policy under different environment dynamics and/or learning new tasks. This is because the estimation of preferences which are only defined in terms of states is less sensitive to the specific environment dynamics in the expert's demonstration dataset. Moreover, in application such as healthcare [39] and autonomous driving [40], where simply imitating the expert policy can potentially result in poor performance, since the learner and the expert may have different transition dynamics. Similar points have also been argued in recent works [14, 41–43].

Next, let us briefly discuss how we can understand (ML-IRL) and Alg.1, when the reward is parameterized as a state-only function. First, it turns out that there is an equivalent formulation of (ML-IRL), when the expert trajectories only contain the visited states.

**Lemma 3.** *Suppose the expert trajectories $\tau$ is sampled from a policy $\pi^{\mathrm{E}}$, and the reward is parameterized as a state-only function $r(s; \theta)$. Then ML-IRL is equivalent to the following:*

$$\min_{\theta} \quad \mathbb{E}_{s_0 \sim \eta(\cdot)}\left[V_{r_\theta, \pi_\theta}^{soft}(s_0)\right] - \mathbb{E}_{s_0 \sim \eta(\cdot)}\left[V_{r_\theta, \pi^E}^{soft}(s_0)\right] \tag{12a}$$

$$s.t. \quad \pi_\theta := \arg\max_{\pi} \mathbb{E}_\pi\left[\sum_{t=0}^{\infty} \gamma^t\left(r(s_t; \theta) + \mathcal{H}(\pi(\cdot|s_t))\right)\right]. \tag{12b}$$

Please see Appendix F for the detailed derivation. Intuitively, the above lemma says that, when dealing with the state-only IRL, (ML-IRL) minimizes the gap between the soft value functions of the optimal policy $\pi_\theta$ and that of the expert policy $\pi^{\mathrm{E}}$. Moreover, Alg.1 can also be easily implemented with the state-only reward. In fact, the entire algorithm essentially stays the same, and the only change is that $r(s, a; \theta)$ will be replaced by $r(s; \theta)$. In this way, by only using the visited states in the trajectories, one can still compute the stochastic gradient estimator in (7). Therefore, even under the state-only IRL setting where the expert dataset only contains visited states, our formulation and the proposed algorithm still work if we parameterize the reward as a state-only function.

## 7 Numerical Results

In this section, we test the performance of our algorithm on a diverse collection of RL tasks and environments. In each experiment set, we train algorithms until convergence and average the scores of the trajectories over multiple random seeds. The hyperparameter settings and simulation details are provided in Appendix B.

**MuJoCo Tasks For Inverse Reinforcement Learning.** In this experiment set, we test the performance of our algorithm on imitating the expert behavior. We consider several high-dimensional robotics control tasks in MuJoCo [44]. Two class of existing algorithms are considered as the comparison baselines: 1) imitation learning algorithms that only learn the policy to imitate the expert, including Behavior Cloning (BC) [45] and Generative Adversarial Imitation Learning (GAIL) [10]; 2) IRL algorithms which learn a reward function *and* a policy simultaneously, including Adversarial Inverse Reinforcement Learning (AIRL) [11], $f$-IRL [14] and IQ-Learn [2]. To ensure fair comparison, all imitation learning / IRL algorithms use soft Actor-Critic [22] as the base RL algorithm. For the expert dataset, we use the data provided in the official implementation[2] of $f$-IRL.

In this experiment, we implement two versions of our proposed algorithm: ML-IRL(State-Action) where the reward is parameterized as a function of state and action; ML-IRL(State-Only) which utilizes the state-only reward function. In Table 1, we present the simulation results under a limited data regime where the expert dataset only contains a single expert trajectory. The scores (cumulative rewards) reported in the table is averaged over 6 random seeds. In each random seed, we train algorithm from initialization and collect 20 trajectories to average their cumulative rewards after the algorithms converge. The results reported in Table 1 show that our proposed algorithms outperform the baselines. The numerical results with confidence intervals are in Table 3 (See Appendix).

We observe that BC fails to imitate the expert's behavior. It is due to the fact that BC is based on supervised learning and thus could not learn a good policy under such a limited data regime. Moreover, we notice the training of IQ-Learn is unstable, which may be due to its inaccurate approximation to the soft Q-function. Therefore, in the MuJoCo tasks where IQ-Learn does not perform well, so that we cannot match the results presented in the original paper [2], we directly report results from there (and mark them by $*$ in Table 1). The results of AIRL are not presented in Table 1 since it performs poorly even after spending significant efforts in parameter tuning (similar observations have been made in in [46, 14]).

**Transfer Learning Across Changing Dynamics.** We further evaluate IRL algorithms on the transfer learning setting. We follows the environment setup in [11], where two environments with different dynamics are considered: `Custom-Ant` vs `Disabled-Ant`. We compare ML-IRL(State-Only) with several existing IRL methods: 1) AIRL [11], 2) $f$-IRL [14]; 3) IQ-Learn [2].

---

[2] https://github.com/twni2016/f-IRL

| Task | BC | GAIL | IQ-Learn | $f$-IRL | ML-IRL (State-Only) | ML-IRL (State-Action) | Expert |
|---|---|---|---|---|---|---|---|
| Hopper | 20.49 | 2815.59 | 2981.01 | 3074.55 | 3089.79 | **3121.68** | 3592.63 |
| Half-Cheetah | -1.87 | 3301.52 | 4175.88 | 4375.88 | **4472.85** | 4086.92 | 5098.3 |
| Walker | -14.01 | 1112.79 | 3961.42 | 4464.20 | 4380.17 | **4504.88** | 5344.21 |
| Ant | 760.46 | 1154.27 | 4362.90* | 4571.71 | 4675.34 | **4984.34** | 5926.18 |
| Humanoid | 78.48 | 3016.40 | 5227.10* | 5243.90 | **5390.31** | 5240.57 | 5351.08 |

Table 1: **MuJoCo Results.** The performance of benchmark algorithms under a single expert trajectory.

We consider two transfer learning settings: 1) data transfer; 2) reward transfer. For both settings, the expert dataset / trajectories are generated in `Custom-Ant`. In the data transfer setting, we train IRL agents in `Disabled-Ant` by using the expert trajectories, which are generated in `Custom-Ant`. In the reward transfer setting, we first use IRL algorithms to infer the reward functions in `Custom-Ant`, and then transfer these recovered reward functions to `Disabled-Ant` for further evaluation. In both settings, we also train SAC with the ground-truth reward in `Disabled-Ant` and report the scores.

The numerical results are reoprted in Table 2. the proposed ML-IRL(State-Only) achieves superior performance compared with the existing IRL benchmarks in both settings. We notice that IQ-Learn fails in both settings since it indirectly recovers the reward function from a soft Q-function approximator, which could be inaccurate and is highly dependent upon the environment dynamics. Therefore, the reward function recovered by IQ-Learn can not be disentangled from the expert actions and environment dynamics, which leads to its failures in the transfer learning tasks.

| Setting | IQ-Learn | AIRL | $f$-IRL | ML-IRL(State-Only) | Groud-Truth |
|---|---|---|---|---|---|
| Data Transfer | -11.78 | -5.39 | 188.85 | **221.51** | 320.15 |
| Reward Transfer | -1.04 | 130.3 | 156.45 | **187.69** | 320.15 |

Table 2: **Transfer Learning.** The performance of benchmark algorithms under a single expert trajectory. The scores in the table are obtained similarly as in Table 1.

# 8 Conclusion

In this paper, we present a maximum likelihood IRL formulation and propose a provably efficient algorithm with a single-loop structure. To our knowledge, we provide the first non-asymptotic analysis for IRL algorithm under nonlinear reward parameterization. As a by-product, when we parameterize the reward as a state-only function, our algorithm could work in state-only IRL setting and enable reward transfer to new environments with different dynamics. Our algorithm outperforms existing IRL methods on high-dimensional robotics control tasks and corresponding transfer learning settings. A limitation of our method is the requirement for online training, so one future direction of this work is to further extend our algorithm and the theoretical analysis to the offline IRL setting.

## Potential Negative Social Impacts

Since IRL methods aim to recover the reward function and the associated optimal policy from the observed expert dataset, potential negative social impacts may occur if there are bad demonstrations included in the expert dataset. Thus, for sensitive applications such as autonomous driving and clinical decision support, additional care should be taken to avoid negative biases from the expert demonstrations and ensure safe adaptation.

## Acknowledgments

We thank the anonymous reviewers for their valuable comments. M. Hong and S. Zeng are partially supported by NSF grants CIF-1910385, CMMI-1727757, and AFOSR grant 19RT0424. A. Garcia would like to acknowledge partial support from grant FA9550-19-1-00347 by AFOSR.

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
