# Appendix

## A   Related Works

Under the maximum entropy framework, IRL algorithms [7, 9] are proposed to learn the nonlinear structure of reward function. In general, these works [5, 7, 9] recover the reward function through minimizing forward KL divergence in trajectory space. There is one branch of IRL methods which train a generative adversial netowrk (GAN) [13] to learn the reward function through designing a specific structure in the discriminator network. In [8], the authors reveal a connection between GAN and guided cost learning (GCL) [9]. Observing the Lagrangian function of the MaxEnt-IRL [5, 18, 20] is a min-max problem with convex-nonconcave structure, [12, 37] first swap the optimization order of the reward parameter and the policy parameter and further regularize the reward parameter for analyzing the convergence of such constructed nonconcave strongly-convex optimization problem. In [47], in order to provide a non-asymptotic analysis of the IRL problem, the authors introduce a bilinear saddlepoint framework through using Lagrangian duality.

Under the MaxEnt-IRL framework, there is a line of works focusing on disentangling the reward function from the environment dynamics, so that the recovered reward functions could be transferred across environments with different dynamics. In [11], the authors propose an algorithm called adversarial inverse reinforcement learning (AIRL). Constructing the estimated reward as a function which depends on the current state and next state, AIRL enables the agent to learn policies in a new environment through leveraging the estimated reward function recovered from the training environment. In [4], the authors prove necessary and sufficient conditions for reward identifiability in deterministic MDP models with the maximum entropy reinforcement learning objective. In [48], the authors present a theoretical analysis to show the necessary and sufficient condition to identify an action-independent time-homogeneous reward function under MaxEnt-IRL.

A recent line of works consider a more challenging setting, where the learner has no access to the expert environment, and there is a transition dynamics mismatch between the expert and the learner. In [46], the authors propose a state alignment based imitation learning method so that the imitator could follow the state sequences in expert demonstrations as much as possible. Arguing that the expert actions are not efficient demonstrations under transition dynamics mismatch, [41] further develops a state-only imitation learning method. In [42], the authors revisit the Maximum Causal Entropy IRL when there is a transition dynamics mismatch between the expert and the learner. A theoretical analysis is further provided in [42] to show the upper bound on the learner's performance degradation, which is measured in terms of the $\ell_1$-distance between the transition dynamics of the expert and the learner.

We would like to further introduce several interesting works which seek to make imitating expert policy more tractable. In [49], the authors utilize random network distillation and propose a new general framework of imitation learning via expert policy support estimation. In [50], a ranking-based imitation learning method is proposed and the authors show that such imitation learning method could outperform the demonstrator. Witnessing the instability of adversarial training, [51] proposes an imitation learning method without performing any policy optimization steps.

In the end, we introduce the wide applications of inverse reinforcement learning. The problem of inverse reinforcement learning IRL has been widely in studied by the robotics and artificial intelligence research communities [52, 53]. It has also been applied (under the label of dynamic discrete choice estimation) in a wide variety of application domains including modeling of employee retirement decisions [54], occupational choices and career decisions of young professionals [55], incentives to get teachers to work [56], adult women's mammography decisions [57], trade and labor markets [58], car ownership [59]. The techniques developed in the present paper will enable new applications to settings with high dimensional state space.

## B   Experiment Details

### B.1   MuJoCo Tasks For Inverse Reinforcement Learning.

In all experiments, we test the performance of benchmark algorithms on `Hopper`, `Half-Cheetah`, `Walker`, `Ant`, `Humanoid` environments from OpenAI Gym. To ensure fair comparison, we use

**Algorithm 2** *Practical Implementation of ML-IRL*

---

**Input:** Initialize reward parameter $\theta_0$ and policy $\pi_0$. Set the reward parameter's stepsize as $\alpha$.
**Data Preparation:** Collect a dataset $\mathcal{D}$ which contains multiple expert trajectories
**for** $k = 0, 1, \ldots, K - 1$ **do**
    **Policy Update:** $\pi_{k+1} \leftarrow$ several SAC steps under reward function $r(\cdot, \cdot; \theta_k)$ and policy $\pi_k$.
    **Data Sampling I:** Sampling expert trajectory $\tau_k^{\text{E}} := \{s_t, a_t\}_{t \geq 0}$ from the dataset $\mathcal{D}$
    **Data Sampling II:** Sampling agent trajectory $\tau_k^{\text{A}} := \{s_t, a_t\}_{t \geq 0}$ from the policy $\pi_{k+1}$
    **Estimating Gradient:** $g_k := h(\theta_k; \tau_k^{\text{E}}) - h(\theta_k; \tau_k^{\text{A}})$ where $h(\theta; \tau) := \sum_{t \geq 0} \gamma^t \nabla_\theta r(s_t, a_t; \theta)$
    **Reward Parameter Update:** $\theta_{k+1} := \theta_k + \alpha g_k$
**end for**

---

an open-source implementation[3] of SAC as the base RL algorithm for all imitation learning / IRL methods. Moreover, Adam is used as the optimizer in SAC.

In SAC, both policy network and Q-network are $(64, 64)$ MLPs with ReLU activation function, and we set their stepsizes as $3 \times 10^{-3}$. Moreover, in our proposed algorithms, we parameterize the reward function by a $(64, 64)$ MLPs with ReLU activation function. For the reward network, we use Adam as the optimizer and the stepsize is set to be $1 \times 10^{-4}$.

We present the practical implementation procedure of our proposed algorithm in Table.2. At each iteration, we first warm-start both policy network and Q-network in SAC by using the trained neural networks from the previous iteration. Then, we run 10 episodes in the corresponding MuJoCo environment to train the policy network and Q-network in SAC. After that, we sample 5 agent trajectories and expert trajectories to construct the reward gradient estimator, and then update the reward network by a gradient update.

For the imitation learning / IRL benchmark algorithms, we use their open-source implementations in our experiments. The official implementations of $f$-IRL is provided in `https://github.com/twni2016/f-IRL`. The offical code base for IQ-Learn is provided in `https://github.com/Div99/IQ-Learn`. For the remaining benchmarks including BC, GAIL and AIRL, we refer to a open-source implementation: `https://github.com/KamyarGh/rl_swiss`.

## B.2 Transfer Learning Across Changing Dynamics.

In this experiment, we follow the setup in [11]. A standard ant (`Custom-Ant`) and an ant with two disabled legs (`Disabled-Ant`) are simulated in MuJoCo. For all benchmark algorithms tested in this experiment, we follow same network structure and hyperparameter settings described in Section B.1.

---

[3]`https://github.com/openai/spinningup`

Here, we provide a supplementary experiment result to show the performance of benchmark algorithms under different number of expert trajectory. The performance of AIRL and IQ-Learn is not presented in Table 3 since we found their training is unstable (as we have mentioned in Sec.7). The scores in Hopper are recorded after $1 \times 10^6$ environment steps and the scores in other environments are recorded after $2 \times 10^6$ environment steps. The scores are reported after 6 independent Monte Carlo (MC) trials for each algorithm.

| Task | Hopper | | |
|---|---|---|---|
| # Expert Trajectory | 1 | 5 | 10 |
| Expert Performance | 3592.63 | $3530.63 \pm 2.73$ | $3531.72 \pm 6.41$ |
| BC | $20.49 \pm 3.24$ | $104.50 \pm 59.12$ | $378.42 \pm 56.08$ |
| GAIL | $2815.59 \pm 203.80$ | $2840.71 \pm 166.36$ | $2941.82 \pm 128.34$ |
| $f$-IRL | $3074.55 \pm 237.03$ | $3118.49 \pm 83.25$ | $\mathbf{3127.05} \pm 103.60$ |
| ML-IRL(State-Only) | $3089.79 \pm 18.39$ | $3116.55 \pm 49.54$ | $3039.91 \pm 50.61$ |
| ML-IRL(State-Action) | $\mathbf{3121.68} \pm 286.58$ | $\mathbf{3200.33} \pm 114.28$ | $2943.42 \pm 271.20$ |

| Task | Half-Cheetah | | |
|---|---|---|---|
| # Expert Trajectory | 1 | 5 | 10 |
| Expert Performance | 5098.30 | $5072.53 \pm 145.12$ | $5043.02 \pm 104.32$ |
| BC | $-1.87 \pm 0.24$ | $145.72 \pm 78.45$ | $342.64 \pm 154.97$ |
| GAIL | $3301.52 \pm 243.79$ | $3465 \pm 178.44$ | $3400.05 \pm 115.92$ |
| $f$-IRL | $4375.88 \pm 230.41$ | $4438.96 \pm 365.20$ | $4427.91 \pm 426.64$ |
| ML-IRL(State-Only) | $\mathbf{4472.85} \pm 166.61$ | $\mathbf{4669.85} \pm 483.51$ | $\mathbf{4653.11} \pm 401.04$ |
| ML-IRL(State-Action) | $4086.92 \pm 195.30$ | $4338.35 \pm 107.07$ | $4416.79 \pm 61.35$ |

| Task | Walker | | |
|---|---|---|---|
| # Expert Trajectory | 1 | 5 | 10 |
| Expert Performance | 5344.21 | $5471.58 \pm 13.93$ | $5471.70 \pm 10.59$ |
| BC | $-14.01 \pm 0.24$ | $285.43 \pm 3.24$ | $450.71 \pm 72.08$ |
| GAIL | $1112.79 \pm 143.81$ | $3523.42 \pm 487.70$ | $3789.40 \pm 61.62$ |
| $f$-IRL | $4464.20 \pm 332.49$ | $\mathbf{4507.58} \pm 254.13$ | $4164.77 \pm 648.66$ |
| ML-IRL(State-Only) | $4380.17 \pm 393.90$ | $4402.41 \pm 491.72$ | $\mathbf{4541.04} \pm 272.54$ |
| ML-IRL(State-Action) | $\mathbf{4504.88} \pm 120.82$ | $4027.75 \pm 478.39$ | $4114.00 \pm 732.54$ |

| Task | Ant | | |
|---|---|---|---|
| # Expert Trajectory | 1 | 5 | 10 |
| Expert Performance | 5926.18 | $5856.84 \pm 79.48$ | $5901.73 \pm 122.40$ |
| BC | $760.46 \pm 0.59$ | $942.58 \pm 14.87$ | $1032.70 \pm 73.94$ |
| GAIL | $1154.27 \pm 422.23$ | $3042.55 \pm 403.72$ | $4233.51 \pm 280.51$ |
| $f$-IRL | $4571.71 \pm 169.97$ | $4681.54 \pm 221.83$ | $4574.20 \pm 303.49$ |
| ML-IRL(State-Only) | $4675.34 \pm 26.03$ | $4632.38 \pm 77.90$ | $4498.31 \pm 212.81$ |
| ML-IRL(State-Action) | $\mathbf{4984.34} \pm 177.97$ | $\mathbf{5190.48} \pm 159.65$ | $\mathbf{5011.67} \pm 252.75$ |

| Task | Humanoid | | |
|---|---|---|---|
| # Expert Trajectory | 1 | 5 | 10 |
| Expert Performance | 5351.08 | $5339.12 \pm 22.21$ | $5343.76 \pm 24.19$ |
| BC | $78.48 \pm 0.34$ | $532.61 \pm 48.74$ | $582.78 \pm 18.82$ |
| GAIL | $3016.40 \pm 714.26$ | $3030.25 \pm 594.38$ | $3204.10 \pm 358.88$ |
| $f$-IRL | $5243.90 \pm 163.64$ | $5338.20 \pm 122.219$ | $\mathbf{5472.72} \pm 117.21$ |
| ML-IRL(State-Only) | $\mathbf{5390.31} \pm 161.98$ | $\mathbf{5431.23} \pm 244.17$ | $5257.24 \pm 323.61$ |
| ML-IRL(State-Action) | $5240.57 \pm 72.18$ | $5251.86 \pm 218.98$ | $5342.98 \pm 608.32$ |

Table 3: **MuJoCo Results.** The performance versus different number of expert trajectory.

## C Auxiliary Lemmas

Throughout this section, we assume Assumptions 1 - 2 hold true.

**Lemma 4.** *([60, Lemma 3]) Consider the initialization distribution $\eta(\cdot)$ and transition kernel $\mathcal{P}(\cdot|s,a)$. Under $\eta(\cdot)$ and $\mathcal{P}(\cdot|s,a)$, denote $d_w(\cdot, \cdot)$ as the state-action visitation distribution of MDP with the Boltzman policy parameterized by parameter $w$. Suppose Assumption 1 holds, for all policy parameter $w$ and $w'$, we have*

$$\|d_w(\cdot, \cdot) - d_{w'}(\cdot, \cdot)\|_{TV} \leq C_d\|w - w'\| \tag{13}$$

*where $C_d$ is a positive constant.*

**Lemma 5.** *([21, Theorem 4]) Under a reward function $r(\cdot, \cdot)$, given a policy $\pi$, we define a new policy $\tilde{\pi}$ as*

$$\tilde{\pi}(\cdot|s) \propto \exp\left(Q^{soft}_{r,\pi}(s, \cdot)\right), \ \forall s \in \mathcal{S}.$$

*For any $s \in \mathcal{S}, a \in \mathcal{A}$, it holds that $Q^{soft}_{r,\tilde{\pi}}(s, a) \geq Q^{soft}_{r,\pi}(s, a)$.*

Next, in order to facilitate analysis for entropy-regularized MDPs, we introduce a "soft" Bellman optimality operator $\mathcal{T} : \mathbb{R}^{|\mathcal{S}|\times|\mathcal{A}|} \to \mathbb{R}^{|\mathcal{S}|\times|\mathcal{A}|}$ as follows:

$$\mathcal{T}(Q)(s,a) := r(s,a) + \gamma\mathbb{E}_{s'\sim\mathcal{P}(\cdot|s,a)}\left[\max_{\pi(\cdot|s)} \mathbb{E}_{a'\sim\pi(\cdot|s')}\left[Q(s', a') - \log\pi(a'|s')\right]\right]. \tag{14}$$

In the following lemma, the properties of entropy-regularized MDPs are characterized.

**Lemma 6.** *([28, Lemma 2]) The operator $\mathcal{T}$ as defined in (14) satisfies the properties below:*

- *$\mathcal{T}$ has the following closed-form expression:*

$$\mathcal{T}(Q)(s,a) = r(s,a) + \gamma\mathbb{E}_{s'\sim\mathcal{P}(\cdot|s,a)}\left[\log\left(\sum_{a'}\exp\left(Q(s', a')\right)\right)\right]. \tag{15}$$

- *$\mathcal{T}$ is a $\gamma$-contraction in the $\ell_\infty$ norm, namely, for any $Q_1, Q_2 \in \mathbb{R}^{|\mathcal{S}|\times|\mathcal{A}|}$, it holds that*

$$\|\mathcal{T}(Q_1) - \mathcal{T}(Q_2)\|_\infty \leq \gamma\|Q_1 - Q_2\|_\infty. \tag{16}$$

- *Under a given reward function $r(\cdot, \cdot)$, the corresponding optimal soft Q-function $Q^{soft}_{r,\pi^*}$ is a unique fixed point of the operator $\mathcal{T}$, namely,*

$$\mathcal{T}(Q^{soft}_{r,\pi^*}) = Q^{soft}_{r,\pi^*} \tag{17}$$

*Proof.* This Lemma is proved in [28, Lemma]. We refine its analysis as below.

We first show that

$$\mathbb{E}_{a\sim\pi(\cdot|s)}\left[Q(s,a) - \log\pi(a|s)\right] = \sum_a \pi(a|s)\log\left(\frac{\exp(Q(s,a))}{\pi(a|s)}\right) \overset{(i)}{\leq} \log\left(\sum_a \exp\left(Q(s,a)\right)\right) \tag{18}$$

where (i) is from Jensen's inequality. Moreover, the equality between both sides of (i) holds when the policy $\pi$ has the expression $\pi(\cdot|s) \propto \exp(Q(s, \cdot))$. Therefore, through applying the inequality (18) to (14), it obtains that

$$\mathcal{T}(Q)(s,a) = r(s,a) + \gamma\mathbb{E}_{s'\sim\mathcal{P}(\cdot|s,a)}\left[\log\left(\sum_{a'}\exp\left(Q(s', a')\right)\right)\right], \tag{19}$$

which proves the equality (15).

We define $\|Q_1 - Q_2\|_\infty := \max_{s \in \mathcal{S}, a \in \mathcal{A}} |Q_1(s,a) - Q_2(s,a)|$ and $\epsilon = \|Q_1 - Q_2\|_\infty$. Then for any $s \in \mathcal{S}$ and $a \in \mathcal{A}$, it follows that

$$\log \left( \sum_a \exp \left( Q_1(s,a) \right) \right) \leq \log \left( \sum_a \exp \left( Q_2(s,a) + \epsilon \right) \right)$$

$$= \log \left( \exp(\epsilon) \sum_a \exp \left( Q_2(s,a) \right) \right)$$

$$= \epsilon + \log \left( \sum_a \exp \left( Q_2(s,a) \right) \right)$$

Similarly, it is easy to obtain that $\log \left( \sum_a \exp \left( Q_1(s,a) \right) \right) \geq -\epsilon + \log \left( \sum_a \exp \left( Q_2(s,a) \right) \right)$. Hence, it leads to the contraction property that

$$\|\mathcal{T}(Q_1) - \mathcal{T}(Q_2)\|_\infty \leq \gamma \epsilon = \gamma \|Q_1 - Q_2\|_\infty \tag{20}$$

which proves the contraction property (16).

Moreover, we have

$$\mathcal{T}(Q^{\text{soft}}_{r,\pi^*})(s,a) \stackrel{(i)}{=} r(s,a) + \gamma \mathbb{E}_{s' \sim \mathcal{P}(\cdot|s,a)} \left[ \log \left( \sum_{a'} \exp \left( Q^{\text{soft}}_{r,\pi^*}(s',a') \right) \right) \right] \stackrel{(ii)}{=} Q^{\text{soft}}_{r,\pi^*}(s,a) \tag{21}$$

where (i) follows the equality (19). Based on the defition of the soft Q-function $Q^{\text{soft}}_{r,\pi^*}$, we have

$$Q^{\text{soft}}_{r,\pi^*}(s,a) = r(s,a) + \gamma \mathbb{E}_{s' \sim \mathcal{P}(\cdot|s,a)} \left[ \mathbb{E}_{a' \sim \pi^*(\cdot|s')} [-\log \pi^*(a'|s') + Q^{\text{soft}}_{r,\pi^*}(s',a')] \right]. \tag{22}$$

We prove the equality (ii) in (21) through combining (22) and the fact that the optimal soft policy has the closed form $\pi^*(\cdot|s) \propto \exp \left( Q^{\text{soft}}_{r,\pi^*}(s,\cdot) \right)$. Suppose two different fixed points of the soft Bellman operator exist, then it contradicts with the contraction property in (20).

Hence, we proved the uniqueness of the optimal soft $Q$-function $Q^{\text{soft}}_{r,\pi^*}$. Moreover, the optimal soft $Q$-function $Q^{\text{soft}}_{r,\pi^*}$ is a fixed point to the soft Bellman operator $\mathcal{T}$ in (17). $\qquad\square$

**Lemma 7.** *Suppose Assumption 2 holds. Under an arbitrary policy $\pi$, for any $s \in \mathcal{S}$, $a \in \mathcal{A}$ and any reward parameters $\theta_1$ and $\theta_2$, the following inequality holds:*

$$|Q^{soft}_{r_{\theta_1},\pi}(s,a) - Q^{soft}_{r_{\theta_2},\pi}(s,a)| \leq L_q \|\theta_1 - \theta_2\|,$$

*where $L_q := \frac{L_r}{1-\gamma}$ and $L_r$ is the positive constant in Assumption 2.*

*Proof.* Based on the definition of soft-Q function, we have

$$Q^{\text{soft}}_{r,\pi}(s,a) := r(s,a) + \mathbb{E}_\pi \left[ \sum_{t=1}^\infty \gamma^t \left( r(s_t,a_t) + \mathcal{H}(\pi(\cdot|s_t)) \right) \Big| (s_0,a_0) = (s,a) \right].$$

Then it holds that

$$|Q^{\text{soft}}_{r_{\theta_1},\pi}(s,a) - Q^{\text{soft}}_{r_{\theta_2},\pi}(s,a)|$$

$$= \left| \mathbb{E}_\pi \left[ \sum_{t=0}^\infty \gamma^t \left( r(s_t,a_t;\theta_1) - r(s_t,a_t;\theta_2) \right) \right] \right|$$

$$\stackrel{(i)}{\leq} \mathbb{E}_\pi \left[ \sum_{t=0}^\infty \gamma^t \left| r(s_t,a_t;\theta_1) - r(s_t,a_t;\theta_2) \right| \right]$$

$$\stackrel{(ii)}{\leq} \mathbb{E}_\pi \left[ \sum_{t=0}^\infty \gamma^t \left\| \max_\theta \nabla_\theta r(s_t,a_t;\theta) \right\| \cdot \left\| \theta_1 - \theta_2 \right\| \right]$$

$$\stackrel{(iii)}{\leq} \mathbb{E}_\pi \left[ \sum_{t=0}^\infty \gamma^t L_r \|\theta_1 - \theta_2\| \right]$$

$$= \frac{L_r}{1-\gamma} \|\theta_1 - \theta_2\| \tag{23}$$

where (i) follows Jensen's inequality; (ii) follows the mean value theorem; (iii) follows inequality (9a) in Assumption 2.

$\square$

## D   Proof of Lemma 1

*Proof.* First, we are able to express the objective function $L(\theta)$ in (ML-IRL) as below:

$$L(\theta) := \mathbb{E}_{\tau \sim \pi^{\mathrm{E}}} \left[ \sum_{t=0}^{\infty} \gamma^t \log \pi_\theta(a_t|s_t) \right] \overset{(i)}{=} \mathbb{E}_{\tau \sim \pi^{\mathrm{E}}} \left[ \sum_{t=0}^{\infty} \gamma^t \log \left( \frac{\exp\left(Q^{\mathrm{soft}}_{r_\theta, \pi_\theta}(s_t, a_t)\right)}{\sum_a \exp\left(Q^{\mathrm{soft}}_{r_\theta, \pi_\theta}(s_t, a)\right)} \right) \right]$$

where (i) is due to the fact that the optimal policy has the closed form $\pi_\theta(\cdot|s) \propto \exp\left(Q^{\mathrm{soft}}_{r_\theta, \pi_\theta}(s, \cdot)\right)$. Therefore, we could express the objective function in this form:

$$L(\theta) := \mathbb{E}_{\tau \sim \pi^{\mathrm{E}}} \left[ \sum_{t=0}^{\infty} \gamma^t \left( Q^{\mathrm{soft}}_{r_\theta, \pi_\theta}(s_t, a_t) - \log\left( \sum_a \exp\left(Q^{\mathrm{soft}}_{r_\theta, \pi_\theta}(s_t, a)\right)\right)\right)\right]$$

$$\overset{(i)}{=} \mathbb{E}_{\tau \sim \pi^{\mathrm{E}}} \left[ \sum_{t=0}^{\infty} \gamma^t \left( Q^{\mathrm{soft}}_{r_\theta, \pi_\theta}(s_t, a_t) - V^{\mathrm{soft}}_{r_\theta, \pi_\theta}(s_t)\right)\right]$$

$$= \mathbb{E}_{\tau \sim \pi^{\mathrm{E}}} \left[ \sum_{t=0}^{\infty} \gamma^t \left( r(s_t, a_t; \theta) + \gamma \mathbb{E}_{s_{t+1} \sim \mathcal{P}(\cdot|s_t, a_t)}\left[V^{\mathrm{soft}}_{r_\theta, \pi_\theta}(s_{t+1})\right] - V^{\mathrm{soft}}_{r_\theta, \pi_\theta}(s_t)\right)\right]$$

$$= \mathbb{E}_{\tau \sim \pi^{\mathrm{E}}} \left[ \sum_{t=0}^{\infty} \gamma^t r(s_t, a_t; \theta)\right] + \mathbb{E}_{\tau \sim \pi^{\mathrm{E}}} \left[ \sum_{t=1}^{\infty} \gamma^t V^{\mathrm{soft}}_{r_\theta, \pi_\theta}(s_t)\right] - \mathbb{E}_{\tau \sim \pi^{\mathrm{E}}} \left[ \sum_{t=0}^{\infty} \gamma^t V^{\mathrm{soft}}_{r_\theta, \pi_\theta}(s_t)\right]$$

$$= \mathbb{E}_{\tau \sim \pi^{\mathrm{E}}} \left[ \sum_{t=0}^{\infty} \gamma^t r(s_t, a_t; \theta)\right] - \mathbb{E}_{s_0 \sim \eta(\cdot)}\left[V^{\mathrm{soft}}_{r_\theta, \pi_\theta}(s_0)\right] \tag{24}$$

$$\overset{(ii)}{=} \mathbb{E}_{\tau \sim \pi^{\mathrm{E}}} \left[ \sum_{t=0}^{\infty} \gamma^t r(s_t, a_t; \theta)\right] - \mathbb{E}_{s_0 \sim \eta(\cdot)}\left[\log\left( \sum_a \exp\left(Q^{\mathrm{soft}}_{r_\theta, \pi_\theta}(s_0, a)\right)\right)\right] \tag{25}$$

where (i) and (ii) follows the fact that the the optimal soft value function could be expressed as $V^{\mathrm{soft}}_{r_\theta, \pi_\theta}(s) = \log\left( \sum_a \exp\left(Q^{\mathrm{soft}}_{r_\theta, \pi_\theta}(s, a)\right)\right)$.

Based on (25), we calculate the exact gradient of the objective function $L(\theta)$ as below:

$$\nabla L(\theta) := \mathbb{E}_{\tau \sim \pi^{\mathrm{E}}} \left[ \sum_{t=0}^{\infty} \gamma^t \nabla_\theta r(s_t, a_t; \theta)\right] - \mathbb{E}_{s_0 \sim \eta(\cdot)}\left[\nabla_\theta \log\left( \sum_a \exp\left(Q^{\mathrm{soft}}_{r_\theta, \pi_\theta}(s_0, a)\right)\right)\right]$$

$$= \mathbb{E}_{\tau \sim \pi^{\mathrm{E}}} \left[ \sum_{t=0}^{\infty} \gamma^t \nabla_\theta r(s_t, a_t; \theta)\right] - \mathbb{E}_{s_0 \sim \eta(\cdot)}\left[\sum_a \left( \frac{\exp\left(Q^{\mathrm{soft}}_{r_\theta, \pi_\theta}(s_0, a)\right)}{\sum_{\tilde{a}} \exp\left(Q^{\mathrm{soft}}_{r_\theta, \pi_\theta}(s_0, \tilde{a})\right)} \nabla_\theta Q^{\mathrm{soft}}_{r_\theta, \pi_\theta}(s_0, a)\right)\right]$$

$$= \mathbb{E}_{\tau \sim \pi^{\mathrm{E}}} \left[ \sum_{k=0}^{\infty} \gamma^t \nabla_\theta r(s_t, a_t; \theta)\right] - \mathbb{E}_{s_0 \sim \eta(\cdot)}\left[\sum_a \pi_\theta(a|s_0) \nabla_\theta Q^{\mathrm{soft}}_{r_\theta, \pi_\theta}(s_0, a)\right] \tag{26}$$

Then we need to calculate the gradient $\nabla_\theta Q^{\text{soft}}_{r_\theta,\pi_\theta}(s_0,a_0)$ as follows.

$$\nabla_\theta Q^{\text{soft}}_{r_\theta,\pi_\theta}(s_0,a_0)$$

$$\overset{(i)}{=} \nabla_\theta\left(r(s_0,a_0;\theta) + \gamma\mathbb{E}_{s_1\sim\mathcal{P}(\cdot|s_0,a_0)}\left[V^{\text{soft}}_{r_\theta,\pi_\theta}(s_1)\right]\right)$$

$$\overset{(ii)}{=} \nabla_\theta r(s_0,a_0;\theta) + \gamma\mathbb{E}_{s_1\sim\mathcal{P}(\cdot|s_0,a_0)}\left[\nabla_\theta\log\left(\sum_a\exp\left(Q^{\text{soft}}_{r_\theta,\pi_\theta}(s_0,a)\right)\right)\right]$$

$$= \nabla_\theta r(s_0,a_0;\theta) + \gamma\mathbb{E}_{s_1\sim\mathcal{P}(\cdot|s_0,a_0)}\left[\sum_a\frac{\exp(Q^{\text{soft}}_{r_\theta,\pi_\theta}(s_1,a))}{\sum_{\tilde{a}}\exp(Q^{\text{soft}}_{r_\theta,\pi_\theta}(s_1,\tilde{a}))}\nabla_\theta Q^{\text{soft}}_{r_\theta,\pi_\theta}(s_1,a)\right]$$

$$\overset{(iii)}{=} \nabla_\theta r(s_0,a_0;\theta) + \gamma\mathbb{E}_{s_1\sim\mathcal{P}(\cdot|s_0,a_0)}\left[\sum_a\pi_\theta(a|s_1)\nabla_\theta Q^{\text{soft}}_{r_\theta,\pi_\theta}(s_1,a)\right]$$

$$\overset{(iv)}{=} \nabla_\theta r(s_0,a_0;\theta) + \gamma\mathbb{E}_{s_1\sim\mathcal{P}(\cdot|s_0,a_0),a_1\sim\pi_\theta(\cdot|s_1)}\left[\nabla_\theta\left(r(s_1,a_1;\theta) + \gamma\mathbb{E}_{s_2\sim\mathcal{P}(\cdot|s_1,a_1)}\left[V^{\text{soft}}_{r_\theta,\pi_\theta}(s_2)\right]\right)\right]$$

$$\overset{(v)}{=} \mathbb{E}_{\tau\sim\pi_\theta}\left[\sum_{t\geq0}\nabla_\theta r(s_t,a_t;\theta) \mid s_0,a_0\right] \tag{27}$$

where (i) and (iv) follows the definition of the soft Q-function; (ii) follows the fact that $V^{\text{soft}}_{r_\theta,\pi_\theta}(s) = \log(\sum_a\exp(Q^{\text{soft}}_{r_\theta,\pi_\theta}(s,a)))$; (iii) follows the fact that $\pi_\theta(a|s)\propto\exp(Q^{\text{soft}}_{r_\theta,\pi_\theta}(s,a))$; (v) is shown by recursively applying (i) - (iv).

Finally, plugging equation (27) into (26), the gradient of the maximum likelihood objective is:

$$\nabla L(\theta) = \mathbb{E}_{\tau\sim\pi^{\text{E}}}\left[\sum_{t\geq0}\gamma^t\nabla_\theta r(s_t,a_t;\theta)\right] - \mathbb{E}_{\tau\sim\pi_\theta}\left[\sum_{t\geq0}\gamma^t\nabla_\theta r(s_t,a_t;\theta)\right]. \tag{28}$$

$\square$

# E  Proof of Lemma 2

To proof Lemma 2, we proof the equality (10a) and the equality (10b) respectively. The constants $L_q$ and $L_c$ in Lemma 2 has the expression:

$$L_q := \frac{L_r}{1-\gamma}, \quad L_c := \frac{2L_qL_rC_d\sqrt{|\mathcal{S}|\cdot|\mathcal{A}|}}{1-\gamma} + \frac{2L_g}{1-\gamma}.$$

## E.1  Proof of Inequality (10a)

In this subsection, we prove the inequality (10a) in Lemma 2.

*Proof.* We show that $Q^{\text{soft}}_{r_\theta,\pi_\theta}$ has bounded gradient with respect to any reward parameter $\theta$, then the inequality (10a) holds due to the mean value theorem. According to the equality (27), we have shown the explicit expression of $\nabla_\theta Q^{\text{soft}}_{r_\theta,\pi_\theta}(s,a)$ for any $s\in\mathcal{S}$ and $a\in\mathcal{A}$. Using this expression, we have the following series of relations:

$$\|\nabla_\theta Q^{\text{soft}}_{r_\theta,\pi_\theta}(s,a)\| \overset{(i)}{=} \left\|\mathbb{E}_{\tau\sim\pi_\theta}\left[\sum_{t\geq0}\gamma^t\nabla_\theta r(s_t,a_t;\theta)\,\middle|\,(s_0,a_0)=(s,a)\right]\right\|$$

$$\overset{(ii)}{\leq} \mathbb{E}_{\tau\sim\pi_\theta}\left[\sum_{t\geq0}\gamma^t\left\|\nabla_\theta r(s_t,a_t;\theta)\right\|\,\middle|\,(s_0,a_0)=(s,a)\right]$$

$$\overset{(iii)}{\leq} \mathbb{E}_{\tau\sim\pi_\theta}\left[\sum_{t\geq0}\gamma^tL_r\,\middle|\,(s_0,a_0)=(s,a)\right]$$

$$= \frac{L_r}{1-\gamma} \tag{29}$$

where (i) is from the equality (27) in the proof of Lemma 1, (ii) follows Jensen's inequality and (iii) follows the inequality (9a) in Assumption 2. To complete this proof, we use the mean value theorem to show that

$$|Q_{r_{\theta_1},\pi_{\theta_1}}^{\text{soft}}(s,a) - Q_{r_{\theta_2},\pi_{\theta_2}}^{\text{soft}}(s,a)| \leq \|\max_{\theta} \nabla_\theta Q_{r_\theta,\pi_\theta}^{\text{soft}}(s,a)\| \cdot \|\theta_1 - \theta_2\| \leq L_q \|\theta_1 - \theta_2\|$$

where the last inequality follows (29) and we denote $L_q := \frac{L_r}{1-\gamma}$. Therefore, we have proved the Lipschitz continuous inequality in (10a).

$\square$

### E.2 Proof of Inequality (10b)

In this section, we prove the inequality (10b) in Lemma 2.

*Proof.* According to Lemma 1, the gradient $\nabla L(\theta)$ is expressed as:

$$\nabla L(\theta) = \mathbb{E}_{\tau\sim\pi^{\text{E}}}\left[\sum_{t\geq 0}\gamma^t \nabla_\theta r(s_t,a_t;\theta)\right] - \mathbb{E}_{\tau\sim\pi_\theta}\left[\sum_{t\geq 0}\gamma^t \nabla_\theta r(s_t,a_t;\theta)\right]. \tag{30}$$

Using the above relation, we have

$$\|\nabla L(\theta_1) - \nabla L(\theta_2)\|$$
$$\overset{(i)}{=} \left\|\left(\mathbb{E}_{\tau\sim\pi^{\text{E}}}\left[\sum_{t\geq 0}\gamma^t \nabla_\theta r(s_t,a_t;\theta_1)\right] - \mathbb{E}_{\tau\sim\pi_{\theta_1}}\left[\sum_{t\geq 0}\gamma^t \nabla_\theta r(s_t,a_t;\theta_1)\right]\right) -\right.$$
$$\left.\left(\mathbb{E}_{\tau\sim\pi^{\text{E}}}\left[\sum_{t\geq 0}\gamma^t \nabla_\theta r(s_t,a_t;\theta_2)\right] - \mathbb{E}_{\tau\sim\pi_{\theta_2}}\left[\sum_{t\geq 0}\gamma^t \nabla_\theta r(s_t,a_t;\theta_2)\right]\right)\right\|$$
$$\leq \underbrace{\left\|\mathbb{E}_{\tau\sim\pi^{\text{E}}}\left[\sum_{t\geq 0}\gamma^t \nabla_\theta r(s_t,a_t;\theta_1)\right] - \mathbb{E}_{\tau\sim\pi^{\text{E}}}\left[\sum_{t\geq 0}\gamma^t \nabla_\theta r(s_t,a_t;\theta_2)\right]\right\|}_{:=\text{term A}} +$$
$$\underbrace{\left\|\mathbb{E}_{\tau\sim\pi_{\theta_1}}\left[\sum_{t\geq 0}\gamma^t \nabla_\theta r(s_t,a_t;\theta_1)\right] - \mathbb{E}_{\tau\sim\pi_{\theta_2}}\left[\sum_{t\geq 0}\gamma^t \nabla_\theta r(s_t,a_t;\theta_2)\right]\right\|}_{:=\text{term B}} \tag{31}$$

where (i) follows the exact gradient expression in equation (30). Then we separately analyze term A and term B in (31).

For term A, it follows that

$$\left\|\mathbb{E}_{\tau\sim\pi^{\text{E}}}\left[\sum_{t\geq 0}\gamma^t \nabla_\theta r(s_t,a_t;\theta_1)\right] - \mathbb{E}_{\tau\sim\pi^{\text{E}}}\left[\sum_{t\geq 0}\gamma^t \nabla_\theta r(s_t,a_t;\theta_2)\right]\right\|$$
$$\overset{(i)}{\leq} \mathbb{E}_{\tau\sim\pi^{\text{E}}}\left[\sum_{t\geq 0}\gamma^t \|\nabla_\theta r(s_t,a_t;\theta_1) - \nabla_\theta r(s_t,a_t;\theta_2)\|\right]$$
$$\overset{(ii)}{\leq} \mathbb{E}_{\tau\sim\pi^{\text{E}}}\left[\sum_{t\geq 0}\gamma^t L_g\|\theta_1 - \theta_2\|\right]$$
$$= \frac{L_g}{1-\gamma}\|\theta_1 - \theta_2\| \tag{32}$$

where (i) follows Jensen's inequality and (ii) is from (9b) in Assumption 2.

For the term B, it holds that

$$
\left\| \mathbb{E}_{\tau \sim \pi_{\theta_1}} \left[ \sum_{t \geq 0} \gamma^t \nabla_\theta r(s_t, a_t; \theta_1) \right] - \mathbb{E}_{\tau \sim \pi_{\theta_2}} \left[ \sum_{t \geq 0} \gamma^t \nabla_\theta r(s_t, a_t; \theta_2) \right] \right\|
$$

$$
\overset{(i)}{\leq} \left\| \mathbb{E}_{\tau \sim \pi_{\theta_1}} \left[ \sum_{t \geq 0} \gamma^t \nabla_\theta r(s_t, a_t; \theta_1) \right] - \mathbb{E}_{\tau \sim \pi_{\theta_2}} \left[ \sum_{t \geq 0} \gamma^t \nabla_\theta r(s_t, a_t; \theta_1) \right] \right\|
$$

$$
+ \left\| \mathbb{E}_{\tau \sim \pi_{\theta_2}} \left[ \sum_{t \geq 0} \gamma^t \nabla_\theta r(s_t, a_t; \theta_1) \right] - \mathbb{E}_{\tau \sim \pi_{\theta_2}} \left[ \sum_{t \geq 0} \gamma^t \nabla_\theta r(s_t, a_t; \theta_2) \right] \right\|
$$

$$
\overset{(ii)}{\leq} \frac{1}{1 - \gamma} \left\| \mathbb{E}_{(s,a) \sim d(\cdot, \cdot; \pi_{\theta_1})} \left[ \nabla_\theta r(s_t, a_t; \theta_1) \right] - \mathbb{E}_{(s,a) \sim d(\cdot, \cdot; \pi_{\theta_2})} \left[ \nabla_\theta r(s_t, a_t; \theta_1) \right] \right\|
$$

$$
+ \mathbb{E}_{\tau \sim \pi_{\theta_2}} \left[ \sum_{t \geq 0} \gamma^t \left\| \nabla_\theta r(s_t, a_t; \theta_1) - \nabla_\theta r(s_t, a_t; \theta_2) \right\| \right]
$$

$$
\overset{(iii)}{\leq} \frac{1}{1 - \gamma} \left\| \sum_{s \in \mathcal{S}, a \in \mathcal{A}} \nabla_\theta r(s_t, a_t; \theta_1) \Big( d(s, a; \pi_{\theta_1}) - d(s, a; \pi_{\theta_2}) \Big) \right\| + \mathbb{E}_{\tau \sim \pi_{\theta_2}} \left[ \sum_{k \geq 0} \gamma^k L_g \|\theta_1 - \theta_2\| \right]
$$

$$
\overset{(iv)}{\leq} \frac{2 L_r}{1 - \gamma} \| d(\cdot, \cdot; \pi_{\theta_1}) - d(\cdot, \cdot; \pi_{\theta_1}) \|_{TV} + \frac{L_g}{1 - \gamma} \|\theta_1 - \theta_2\| \tag{33}
$$

where (i) follows the triangle inequality, (ii) is from Jensen's inequality and the definition of the discounted state-action visitation measure $d(s, a; \pi) := (1 - \gamma) \pi(a|s) \sum_{t \geq 0} \gamma^t \mathcal{P}^\pi(s_t = s | s_0 \sim \eta)$; (iii) is from (9b) in Assumption 2;(iv) is from (9a) and the definition of the total variation norm.

Plugging the inequalities (32), (33) to (31), it holds that

$$
\|\nabla L(\theta_1) - \nabla L(\theta_2)\|
$$

$$
\leq \frac{2 L_r}{1 - \gamma} \| d(\cdot, \cdot; \pi_{\theta_1}) - d(\cdot, \cdot; \pi_{\theta_2}) \|_{TV} + \frac{2 L_g}{1 - \gamma} \|\theta_1 - \theta_2\|
$$

$$
\overset{(i)}{\leq} \frac{2 L_r C_d}{1 - \gamma} \| Q^{\text{soft}}_{r_{\theta_1}, \pi_{\theta_1}} - Q^{\text{soft}}_{r_{\theta_2}, \pi_{\theta_2}} \| + \frac{2 L_g}{1 - \gamma} \|\theta_1 - \theta_2\|
$$

$$
\overset{(ii)}{\leq} \frac{2 L_r C_d \sqrt{|\mathcal{S}| \cdot |\mathcal{A}|}}{1 - \gamma} \| Q^{\text{soft}}_{r_{\theta_1}, \pi_{\theta_1}} - Q^{\text{soft}}_{r_{\theta_2}, \pi_{\theta_2}} \|_\infty + \frac{2 L_g}{1 - \gamma} \|\theta_1 - \theta_2\|
$$

$$
\overset{(iii)}{\leq} \left( \frac{2 L_q L_r C_d \sqrt{|\mathcal{S}| \cdot |\mathcal{A}|}}{1 - \gamma} + \frac{2 L_g}{1 - \gamma} \right) \|\theta_1 - \theta_2\|. \tag{34}
$$

Given the fact that $\pi_\theta$ is a Boltzmann policy parameterized by $Q^{\text{soft}}_{r_\theta, \pi_\theta}$ where $\pi_\theta(a|s) \propto \exp(Q^{\text{soft}}_{r_\theta, \pi_\theta}(s, a))$, we show the inequality (i) from the inequality (13) in Lemma 4. Moreover, the inequality (ii) follows the equivalence relation between Frobenius norm and infinity norm and (iii) is from the inequality (10a) in Lemma 2.

Define the constant $L_c := \frac{2 L_q L_r C_d \sqrt{|\mathcal{S}| \cdot |\mathcal{A}|}}{1 - \gamma} + \frac{2 L_g}{1 - \gamma}$, we have the following inequality:

$$
\|\nabla L(\theta_1) - \nabla L(\theta_2)\| \leq L_c \|\theta_1 - \theta_2\|.
$$

Therefore, we complete the proof of the inequality (10b) in Lemma 2. $\qquad \square$

## F    Proof of Lemma 3

*Proof.* Suppose the expert trajectories $\tau$ in ML-IRL is sampled from an expert policy $\pi^{\text{E}}$. Moreover, we parameterize the state-only reward as $r(s; \theta)$. Then the objective function $L(\theta)$ in ML-IRL could

be rewritten as follows.

$$L(\theta) := \mathbb{E}_{\tau \sim \pi^{\mathrm{E}}} \Big[ \sum_{t \geq 0} \gamma^t \log \pi_\theta(a_t|s_t) \Big]$$

$$\overset{(i)}{=} \mathbb{E}_{\tau \sim \pi^{\mathrm{E}}} \Big[ \sum_{t=0}^{\infty} \gamma^t r(s_t; \theta) \Big] - \mathbb{E}_{s_0 \sim \eta(\cdot)} \Big[ V_{r_\theta, \pi_\theta}^{\mathrm{soft}}(s_0) \Big]$$

$$\overset{(ii)}{=} \mathbb{E}_{s_0 \sim \eta(\cdot)} \Big[ V_{r_\theta, \pi^{\mathrm{E}}}^{\mathrm{soft}}(s_0) \Big] - \mathbb{E}_{s_0 \sim \eta(\cdot)} \Big[ V_{r_\theta, \pi_\theta}^{\mathrm{soft}}(s_0) \Big] - H(\pi^{\mathrm{E}}) \tag{35}$$

where (i) follows (24) and the fact that the reward is a state-only function $r(s; \theta)$; (ii) follows the definitions of the soft value function.

Ignoring the constant term $H(\pi^{\mathrm{E}})$ in (35), the maximum likelihood formulation (ML-IRL) is equivalent to the following bi-level problem:

$$\min_\theta \ \mathbb{E}_{s_0 \sim \eta(\cdot)} \big[ V_{r_\theta, \pi_\theta}^{\mathrm{soft}}(s_0) \big] - \mathbb{E}_{s_0 \sim \eta(\cdot)} \big[ V_{r_\theta, \pi^E}^{\mathrm{soft}}(s_0) \big]$$

$$s.t. \ \pi_\theta := \arg\max_\pi \mathbb{E}_\pi \Big[ \sum_{t=0}^{\infty} \gamma^t \Big( r(s_t; \theta) + \mathcal{H}(\pi(\cdot|s_t)) \Big) \Big].$$

Therefore, we complete the proof of Lemma 3. As an alternative interpretation to (ML-IRL), the formulation above aims to minimize the gap between the soft value function of $\pi_\theta$ and $\pi^{\mathrm{E}}$ under the state-only IRL setting. $\qquad \square$

## G   Proof of Theorem 1

*Proof.* Calculate the Lagrangian of MaxEnt-IRL, we obtain that

$$H(\pi) + \left\langle \theta, \mathbb{E}_{\tau \sim \pi}\left[\sum_{t=0}^{\infty}\gamma^t\phi(s_t, a_t)\right] - \mathbb{E}_{\tau \sim \pi^{\mathrm{E}}}\left[\sum_{t=0}^{\infty}\gamma^t\phi(s_t, a_t)\right]\right\rangle + \sum_{s\in\mathcal{S}, t\geq 0} C_{s_t=s}\left(1 - \sum_{a\in\mathcal{A}}\pi(a|s_t)\right)$$

$$= \mathbb{E}_{\tau \sim \pi}\left[\sum_{t=0}^{\infty} -\gamma^t\log\pi(a_t|s_t = s)\right] + \left\langle \theta, \mathbb{E}_{\tau \sim \pi}\left[\sum_{t=0}^{\infty}\gamma^t\phi(s_t, a_t)\right] - \mathbb{E}_{\tau \sim \pi^{\mathrm{E}}}\left[\sum_{t=0}^{\infty}\gamma^t\phi(s_t, a_t)\right]\right\rangle$$

$$+ \sum_{s\in\mathcal{S}, t\geq 0} C_{s_t=s}\left(1 - \sum_{a\in\mathcal{A}}\pi(a|s_t = s)\right) \tag{36}$$

where $\theta$ is the dual variable to ensure the feature matching equality, and $C_{s_t=s}$ is the dual variable to ensure that $\pi$ is a well-defined policy satisfying $\sum_{a\in\mathcal{A}}\pi(a|s_t = s) = 1$.

Then we could calculate the gradient of (36) w.r.t. $\pi(a|s_t = s)$, and set it to 0. Then it holds that

$$0 = \mathcal{P}^\pi(s_t = s)\left(-\gamma^t\big(\log\pi(a|s_t = s) + 1\big) + \mathbb{E}_\pi\left[\sum_{\kappa=t}^{\infty}-\gamma^{\kappa+1}\log\pi(a_{\kappa+1}|s_{\kappa+1}) \mid s_t = s, a_t = a\right]\right.$$

$$\left. + \theta^T\mathbb{E}_\pi\left[\sum_{\kappa=t}^{\infty}\gamma^\kappa\phi(s_\kappa, a_\kappa)|s_t = s, a_t = a\right]\right) - C_{s_t=s}. \tag{37}$$

Dividing $\gamma^t\mathcal{P}^\pi(s_t = s)$ on both sides of (37) and further moving $\log\pi(a|s_t = s)$ to the left side, then we have the equality as below:

$$\log\pi(a|s_t = s) = \left(-\frac{C_{s_t=s}}{\gamma^t\mathcal{P}^\pi(s_t = s)} - 1\right) + \mathbb{E}_\pi\left[\sum_{\kappa=t+1}^{\infty}-\gamma^{\kappa-t}\log\pi(a_\kappa|s_\kappa) \mid s_t = s, a_t = a\right]$$

$$+ \theta^T\mathbb{E}_\pi\left[\sum_{\kappa=t}^{\infty}\gamma^{\kappa-t}\phi(s_\kappa, a_\kappa)|s_t = s, a_t = a\right] \tag{38}$$

Given that $-\frac{C_{s_t=s}}{\gamma^t\mathcal{P}^\pi(s_t=s)} - 1$ is independent of action $a$, we could express the closed form of $\pi(a|s_t = s)$ as below:

$$\pi(a|s_t = s) \propto \exp\left(\mathbb{E}_\pi\left[\sum_{\kappa=t+1}^{\infty}-\gamma^{\kappa-t}\log\pi(a_\kappa|s_\kappa) \mid s_t = s, a_t = a\right] + \theta^T\mathbb{E}_\pi\left[\sum_{\kappa=t}^{\infty}\gamma^{\kappa-t}\phi(s_\kappa, a_\kappa)|s_t = s, a_t = a\right]\right).$$

According to the closed form of the policy above, it shows that $\pi(a|s_t = s)$ is a stationary policy being independent of the time index $t$. Therefore, it holds that $\pi(a|s_t = s) = \pi(a|s)$ for any $t \geq 0$.

Denoting a linearly parameterized reward as $r(s, a; \theta) := \theta^T\phi(s, a)$, it holds that

$$\pi(a|s) \propto \exp\left(\theta^T\mathbb{E}_\pi\left[\sum_{\kappa=0}^{\infty}\gamma^\kappa\phi(s_\kappa, a_\kappa) \mid s_0 = s, a_0 = a\right] + \mathbb{E}_\pi\left[\sum_{\kappa=0}^{\infty}-\gamma^{\kappa+1}\log\pi(a_{\kappa+1}|s_{\kappa+1}) \mid s_0 = s, a_0 = a\right]\right)$$

$$= \exp\left(\mathbb{E}_\pi\left[\sum_{\kappa=0}^{\infty}\gamma^\kappa r(s_\kappa, a_\kappa; \theta) \mid s_0 = s, a_0 = a\right] + \mathbb{E}_\pi\left[\sum_{\kappa=0}^{\infty}-\gamma^{\kappa+1}\log\pi(a_{\kappa+1}|s_{\kappa+1}) \mid s_0 = s, a_0 = a\right]\right) \tag{39}$$

Here, the optimal $\pi(a|s)$ is a function of the dual variables (reward parameters) $\theta$. In the maximum entropy reinforcement learning [21], under a reward function $r(\cdot, \cdot)$ and policy $\pi$, the soft value function and soft Q-function are defined as below:

$$V_{r,\pi}^{\mathrm{soft}}(s) = \mathbb{E}_\pi\left[\sum_{t=0}^{\infty}\gamma^t\Big(r(s_t, a_t) + \mathcal{H}(\pi(\cdot|s_t))\Big)\Big| s_0 = s\right] \tag{40a}$$

$$Q_{r,\pi}^{\mathrm{soft}}(s, a) = r(s, a) + \gamma\mathbb{E}_{s'\sim\mathcal{P}(\cdot|s,a)}\left[V_{r,\pi}^{\mathrm{soft}}(s)\right] \tag{40b}$$

Based on the definitions in (40a) - (40b), we could further express the closed form of the policy in (39) as below:

$$\pi(a|s) = \frac{\exp\left(Q^{\text{soft}}_{r_\theta,\pi}(s,a)\right)}{\sum_{a\in\mathcal{A}}\exp\left(Q^{\text{soft}}_{r_\theta,\pi}(s,a)\right)}. \tag{41}$$

According to [21], under a reward function $r(\cdot,\cdot)$, the optimal soft policy $\pi$ satisfies $\pi(\cdot|s) \propto \exp(Q^{soft}_{r,\pi}(s,\cdot))$. Hence, we have shown that the policy in (41) is the optimal policy under the reward function $r(\cdot,\cdot;\theta)$. After denoting the optimal policy under $r(\cdot,\cdot;\theta)$ as $\pi_\theta$, we have the following relation:

$$\pi_\theta(a|s) = \frac{\exp\left(Q^{\text{soft}}_{r_\theta,\pi_\theta}(s,a)\right)}{\sum_{a\in\mathcal{A}}\exp\left(Q^{\text{soft}}_{r_\theta,\pi_\theta}(s,a)\right)} \stackrel{(a)}{=} \frac{\exp\left(Q^{\text{soft}}_{r_\theta,\pi_\theta}(s,a)\right)}{\exp\left(V^{\text{soft}}_{r_\theta,\pi_\theta}(s)\right)} = \exp\left(Q^{\text{soft}}_{r_\theta,\pi_\theta}(s,a) - V^{\text{soft}}_{r_\theta,\pi_\theta}(s)\right)$$
$$\tag{42}$$

where (a) is due to the equality shown as below:

$$V^{\text{soft}}_{r_\theta,\pi_\theta}(s) = \mathbb{E}_{a\sim\pi_\theta(\cdot|s)}\left[-\log\left(\pi_\theta(a|s)\right) + Q^{\text{soft}}_{r_\theta,\pi_\theta}(s,a)\right]$$

$$= \mathbb{E}_{a\sim\pi_\theta(\cdot|s)}\left[-\log\left(\frac{\exp\left(Q^{\text{soft}}_{r_\theta,\pi_\theta}(s,a)\right)}{\sum_{a\in\mathcal{A}}\exp\left(Q^{\text{soft}}_{r_\theta,\pi_\theta}(s,a)\right)}\right) + Q^{\text{soft}}_{r_\theta,\pi_\theta}(s,a)\right]$$

$$= \log\left(\sum_{a\in\mathcal{A}}\exp\left(Q^{\text{soft}}_{r_\theta,\pi_\theta}(s,a)\right)\right).$$

Rewriting the equality (37), we are able to show the expression of $C_{s_t=s}$ as below:

$$C_{s_t=s} = \mathcal{P}^\pi(s_t=s)\left(-\gamma^t\left(\log\pi_\theta(a|s_t=s)+1\right) + \mathbb{E}_{\pi_\theta}\left[\sum_{\kappa=t}^\infty -\gamma^{\kappa+1}\log\pi_\theta(a_{\kappa+1}|s_{\kappa+1}) \mid s_t=s, a_t=a\right]\right.$$

$$\left. + \theta^T\mathbb{E}_{\pi_\theta}\left[\sum_{\kappa=t}^\infty\gamma^\kappa\phi(s_\kappa,a_\kappa)|s_t=s,a_t=a\right]\right)$$

$$= \gamma^t\mathcal{P}^{\pi_\theta}(s_t=s)\left(-1-\log\pi_\theta(a|s) + \mathbb{E}_{\pi_\theta}\left[\sum_{\kappa=0}^\infty-\gamma^{\kappa+1}\log\pi_\theta(a_{\kappa+1}|s_{\kappa+1}) \mid s_0=s,a_0=a\right]\right.$$

$$\left. + \mathbb{E}_{\pi_\theta}\left[\sum_{\kappa=0}^\infty\gamma^\kappa r(s_\kappa,a_\kappa;\theta)|s_0=s,a_0=a\right]\right)$$

$$\stackrel{(a)}{=} \gamma^t\mathcal{P}^\pi(s_t=s)\left(-1-\log\pi_\theta(a|s)+Q^{\text{soft}}_{r_\theta,\pi_\theta}(s,a)\right)$$

$$\stackrel{(b)}{=} \gamma^t\mathcal{P}^\pi(s_t=s)\left(V^{\text{soft}}_{r_\theta,\pi_\theta}(s)-1\right) \tag{43}$$

where (a) follows the definition of the soft Q-function in (40b), and (b) follows (42). According to (43), we are able to show the exact expression of $C_{s_t=s}$.

Plugging $\pi_\theta$ and $C_{s_t=s}$ into (36), we have

$$\mathbb{E}_{\tau\sim\pi_\theta}\left[\sum_{t=0}^{\infty}-\gamma^t\log\pi_\theta(a_t|s_t)\right]+\left\langle\theta,\mathbb{E}_{\tau\sim\pi_\theta}\left[\sum_{t=0}^{\infty}\gamma^t\phi(s_t,a_t)\right]-\mathbb{E}_{\tau\sim\pi^{\mathrm{E}}}\left[\sum_{t=0}^{\infty}\gamma^t\phi(s_t,a_t)\right]\right\rangle$$

$$+\sum_{s\in\mathcal{S},t\geq0}C_{s_t=s}\left(1-\sum_{a\in\mathcal{A}}\pi_\theta(a|s_t=s)\right)$$

$$\overset{(a)}{=}\mathbb{E}_{\tau\sim\pi_\theta}\left[\sum_{t=0}^{\infty}-\gamma^t\log\pi_\theta(a_t|s_t)\right]+\left\langle\theta,\mathbb{E}_{\tau\sim\pi_\theta}\left[\sum_{t=0}^{\infty}\gamma^t\phi(s_t,a_t)\right]-\mathbb{E}_{\tau\sim\pi^{\mathrm{E}}}\left[\sum_{t=0}^{\infty}\gamma^t\phi(s_t,a_t)\right]\right\rangle$$

$$\overset{(b)}{=}\mathbb{E}_{\tau\sim\pi_\theta}\left[\sum_{t=0}^{\infty}-\gamma^t\left(Q^{\mathrm{soft}}_{r_\theta,\pi_\theta}(s,a)-V^{\mathrm{soft}}_{r_\theta,\pi_\theta}(s)\right)\right]+\left\langle\theta,\mathbb{E}_{\tau\sim\pi_\theta}\left[\sum_{t=0}^{\infty}\gamma^t\phi(s_t,a_t)\right]-\mathbb{E}_{\tau\sim\pi^{\mathrm{E}}}\left[\sum_{t=0}^{\infty}\gamma^t\phi(s_t,a_t)\right]\right\rangle$$

$$\overset{(c)}{=}\mathbb{E}_{\tau\sim\pi_\theta}\left[\sum_{t=0}^{\infty}-\gamma^t\left(\theta^T\phi(s_t,a_t)+\gamma V^{soft}_{r_\theta,\pi_\theta}(s_{t+1})-V^{soft}_{r_\theta,\pi_\theta}(s_t)\right)\right]$$

$$+\left\langle\theta,\mathbb{E}_{\tau\sim\pi_\theta}\left[\sum_{t=0}^{\infty}\gamma^t\phi(s_t,a_t)\right]-\mathbb{E}_{\tau\sim\pi^{\mathrm{E}}}\left[\sum_{t=0}^{\infty}\gamma^t\phi(s_t,a_t)\right]\right\rangle$$

$$=\mathbb{E}_{s_0\sim\eta(\cdot)}\left[V^{\mathrm{soft}}_{r_\theta,\pi_\theta}(s_0)\right]-\theta^T\mathbb{E}_{\tau\sim\pi^{\mathrm{E}}}\left[\sum_{t=0}^{\infty}\gamma^t\phi(s_t,a_t)\right] \tag{44}$$

where (a) is due to the fact that $\sum_{a\in\mathcal{A}}\pi_\theta(a|s_t=s)=1$ for all $a\in\mathcal{A}$ and $s\in\mathcal{S}$; (b) follows (42); (c) is due to the definition of the soft Q-function in (40b). Here, we could further show the problem in (44) is equivalent to ML-IRL as below:

$$\mathbb{E}_{\tau\sim\pi^{\mathrm{E}}}\left[\sum_{t=0}^{\infty}\gamma^t\ln\pi_\theta(a_t|s_t)\right]=\sum_{t=0}^{\infty}\gamma^t\cdot\mathbb{E}_{\tau\sim\pi^{\mathrm{E}}}\left[r(s_t,a_t;\theta)+\gamma V^{\mathrm{soft}}_{r_\theta,\pi_\theta}(s_{t+1})-V^{\mathrm{soft}}_{r_\theta,\pi_\theta}(s_t)\right]$$

$$=\mathbb{E}_{\tau\sim\pi^{\mathrm{E}}}\left[\sum_{t=0}^{\infty}\gamma^t r(s_t,a_t;\theta)\right]+\sum_{t=0}^{\infty}\gamma^t\cdot\mathbb{E}_{\tau\sim\pi^{\mathrm{E}}}\left[\gamma V^{\mathrm{soft}}_{r_\theta,\pi_\theta}(s_{t+1})-V^{\mathrm{soft}}_{r_\theta,\pi_\theta}(s_t)\right]$$

$$=\mathbb{E}_{\tau\sim\pi^{\mathrm{E}}}\left[\sum_{t=0}^{\infty}\gamma^t r(s_t,a_t;\theta)\right]-\mathbb{E}_{s_0\sim\eta(\cdot)}\left[V^{\mathrm{soft}}_{r_\theta,\pi_\theta}(s_0)\right]$$

$$=\theta^T\mathbb{E}_{\tau\sim\pi^{\mathrm{E}}}\left[\sum_{t=0}^{\infty}\gamma^t\phi(s_t,a_t)\right]-\mathbb{E}_{s_0\sim\eta(\cdot)}\left[V^{\mathrm{soft}}_{r_\theta,\pi_\theta}(s_0)\right] \tag{45}$$

Finally, through combining (44) and (45), we are able to know that the maximum likelihood formulation ML-IRL is the dual form of MaxEnt-IRL.

$\square$

# H  Proof of Theorem 2

In this section, we prove (11a) and (11b) respectively, to show the convergence of the lower-level problem and the upper-level problem.

## H.1  Proof of (11a)

*Proof.* In this proof, we first show the convergence of the lower-level variable $\{\pi_k\}_{k\geq 0}$. Recall that we approximate the optimal policy $\pi_{\theta_k}$ by $\pi_{k+1}$ at each iteration $k$. We first analyze the approximation error between $\pi_{\theta_k}$ and $\pi_{k+1}$ as follows. For any $s \in \mathcal{S}$ and $a \in \mathcal{A}$, we have the following relation:

$$
\big| \log\big(\pi_{k+1}(a|s)\big) - \log\big(\pi_{\theta_k}(a|s)\big) \big|
$$
$$
\overset{(i)}{=} \left| \log\left( \frac{\exp\big(Q^{\text{soft}}_{r_{\theta_k},\pi_k}(s,a)\big)}{\sum_{\tilde{a}} \exp\big(Q^{\text{soft}}_{r_{\theta_k},\pi_k}(s,\tilde{a})\big)} \right) - \log\left( \frac{\exp\big(Q^{\text{soft}}_{r_{\theta_k},\pi_{\theta_k}}(s,a)\big)}{\sum_{\tilde{a}} \exp\big(Q^{\text{soft}}_{r_{\theta_k},\pi_{\theta_k}}(s,\tilde{a})\big)} \right) \right|
$$
$$
\overset{(ii)}{\leq} \big|Q^{\text{soft}}_{r_{\theta_k},\pi_k}(s,a) - Q^{\text{soft}}_{r_{\theta_k},\pi_{\theta_k}}(s,a)\big| + \left| \log\left( \sum_{\tilde{a}} \exp\big(Q^{\text{soft}}_{r_{\theta_k},\pi_k}(s,\tilde{a})\big) \right) - \log\left( \sum_{\tilde{a}} \exp\big(Q^{\text{soft}}_{r_{\theta_k},\pi_{\theta_k}}(s,\tilde{a})\big) \right) \right|
$$
(46)

where (i) follows (5) and the fact that $\pi_\theta(a|s) \propto \exp(Q^{\text{soft}}_{r_\theta,\pi_\theta}(s,a))$; (ii) follows the triangle inequality. We further analyze the second term in (46).

We first denote the operator $\log(\|\exp(v)\|_1) := \log(\|\sum_{\tilde{a}\in\mathcal{A}} \exp(v_{\tilde{a}})\|_1)$, where the vector $v \in \mathbb{R}^{|\mathcal{A}|}$ and $v = [v_1, v_2, \cdots, v_{|\mathcal{A}|}]$. Then for any $v', v'' \in \mathbb{R}^{|\mathcal{A}|}$, we have the following relation:

$$
\big| \log\big(\|\exp(v')\|_1\big) - \log\big(\|\exp(v'')\|_1\big) \overset{(i)}{=} \big\langle v' - v'', \nabla_v \log\big(\|\exp(v)\|_1\big)|_{v=v^c} \big\rangle
$$
$$
\leq \|v' - v''\|_\infty \cdot \|\nabla_v \log\big(\|\exp(v)\|_1\big)|_{v=v^c}\|_1
$$
$$
\overset{(ii)}{=} \|v' - v''\|_\infty
$$
(47)

where (i) follows the mean value theorem and $v_c$ is a convex combination of $v'$ and $v''$; (ii) follows the following equalities:

$$
[\nabla_v \log\big(\|\exp(v)\|_1\big)]_i = \frac{\exp(v_i)}{\sum_{1\leq a\leq|\mathcal{A}|} \exp(v_a)}, \quad \|\nabla_v \log\big(\|\exp(v)\|_1\big)\|_1 = 1, \quad \forall v \in \mathbb{R}^{|\mathcal{A}|}.
$$

Through plugging (47) into (46), it holds that

$$
\big| \log\big(\pi_{k+1}(a|s)\big) - \log\big(\pi_{\theta_k}(a|s)\big) \big|
$$
$$
\leq \big|Q^{\text{soft}}_{r_{\theta_k},\pi_k}(s,a) - Q^{\text{soft}}_{r_{\theta_k},\pi_{\theta_k}}(s,a)\big| + \max_{\tilde{a}\in\mathcal{A}} \big|Q^{\text{soft}}_{r_{\theta_k},\pi_k}(s,\tilde{a}) - Q^{\text{soft}}_{r_{\theta_k},\pi_{\theta_k}}(s,\tilde{a})\big|
$$
(48)

Taking the infinity norm over $\mathbb{R}^{|\mathcal{S}|\cdot|\mathcal{A}|}$, the following result holds:

$$
\|\log\pi_{k+1} - \log\pi_{\theta_k}\|_\infty \leq 2\|Q^{\text{soft}}_{r_{\theta_k},\pi_k} - Q^{\text{soft}}_{r_{\theta_k},\pi_{\theta_k}}\|_\infty
$$
(49)

where $\|\log\pi_{k+1} - \log\pi_{\theta_k}\|_\infty = \max_{s\in\mathcal{S},a\in\mathcal{A}} |\log\pi_{k+1}(a|s) - \log\pi_{\theta_k}(a|s)|$ and $\|Q^{\text{soft}}_{r_{\theta_k},\pi_k} - Q^{\text{soft}}_{r_{\theta_k},\pi_{\theta_k}}\|_\infty = \max_{s\in\mathcal{S},a\in\mathcal{A}} |Q^{\text{soft}}_{r_{\theta_k},\pi_k}(s,a) - Q^{\text{soft}}_{r_{\theta_k},\pi_{\theta_k}}(s,a)|$.

Based on the inequality (49), we analyze $\|Q^{\text{soft}}_{r_{\theta_k},\pi_k} - Q^{\text{soft}}_{r_{\theta_k},\pi_{\theta_k}}\|_\infty$ to show the convergence of the policy estimates. It leads to the following analysis:

$$
\|Q^{\text{soft}}_{r_{\theta_k},\pi_k} - Q^{\text{soft}}_{r_{\theta_k},\pi_{\theta_k}}\|_\infty
$$
$$
= \|Q^{\text{soft}}_{r_{\theta_k},\pi_k} - Q^{\text{soft}}_{r_{\theta_k},\pi_{\theta_k}} + Q^{\text{soft}}_{r_{\theta_{k-1}},\pi_{\theta_{k-1}}} - Q^{\text{soft}}_{r_{\theta_{k-1}},\pi_{\theta_{k-1}}} + Q^{\text{soft}}_{r_{\theta_{k-1}},\pi_k} - Q^{\text{soft}}_{r_{\theta_{k-1}},\pi_k}\|_\infty
$$
$$
\leq \|Q^{\text{soft}}_{r_{\theta_k},\pi_{\theta_k}} - Q^{\text{soft}}_{r_{\theta_{k-1}},\pi_{\theta_{k-1}}}\|_\infty + \|Q^{\text{soft}}_{r_{\theta_{k-1}},\pi_k} - Q^{\text{soft}}_{r_{\theta_{k-1}},\pi_{\theta_{k-1}}}\|_\infty + \|Q^{\text{soft}}_{r_{\theta_k},\pi_k} - Q^{\text{soft}}_{r_{\theta_{k-1}},\pi_k}\|_\infty
$$
$$
\overset{(i)}{\leq} L_q\|\theta_k - \theta_{k-1}\| + \|Q^{\text{soft}}_{r_{\theta_{k-1}},\pi_k} - Q^{\text{soft}}_{r_{\theta_{k-1}},\pi_{\theta_{k-1}}}\|_\infty + \|Q^{\text{soft}}_{r_{\theta_k},\pi_k} - Q^{\text{soft}}_{r_{\theta_{k-1}},\pi_k}\|_\infty
$$
$$
\overset{(ii)}{\leq} \|Q^{\text{soft}}_{r_{\theta_{k-1}},\pi_k} - Q^{\text{soft}}_{r_{\theta_{k-1}},\pi_{\theta_{k-1}}}\|_\infty + 2L_q\|\theta_k - \theta_{k-1}\|
$$
(50)

where (i) is from (10a) in Lemma 2; (ii) follows Lemma 7. Based on (50), we further analyze the two terms in (50) as below.

Recall Lemma 6, we have the "soft" Bellman operator expressed as below:

$$\mathcal{T}_\theta(Q)(s,a) = r(s,a;\theta) + \gamma\mathbb{E}_{s'\sim P(\cdot|s',a')}\left[\log\left(\sum_{a'}\exp\left(Q(s',a')\right)\right)\right] \tag{51}$$

According to the soft Bellman operator, it holds that

$$
\begin{aligned}
Q^{\text{soft}}_{r_{\theta_k},\pi_{k+1}}(s,a) &= r(s,a;\theta_k) + \gamma\mathbb{E}_{s'\sim\mathcal{P}(\cdot|s,a)}[V^{\text{soft}}_{r_{\theta_k},\pi_{k+1}}(s')]\\
&= r(s,a;\theta_k) + \gamma\mathbb{E}_{s'\sim\mathcal{P}(\cdot|s,a),a'\sim\pi_{k+1}(\cdot|s')}[-\log\pi_{k+1}(a'|s') + Q^{\text{soft}}_{r_{\theta_k},\pi_{k+1}}(s',a')]\\
&\overset{(i)}{\geq} r(s,a;\theta_k) + \gamma\mathbb{E}_{s'\sim\mathcal{P}(\cdot|s,a),a'\sim\pi_{k+1}(\cdot|s')}[-\log\pi_{k+1}(a'|s') + Q^{\text{soft}}_{r_{\theta_k},\pi_k}(s',a')]\\
&\overset{(ii)}{=} r(s,a;\theta_k) + \gamma\mathbb{E}_{s'\sim\mathcal{P}(\cdot|s,a)}\left[\log\left(\sum_{a'}\exp\left(Q^{\text{soft}}_{r_{\theta_k},\pi_k}(s',a')\right)\right)\right]\\
&\overset{(iii)}{=} \mathcal{T}_{\theta_k}(Q^{\text{soft}}_{r_{\theta_k},\pi_k})(s,a)
\end{aligned}
\tag{52}
$$

where (i) follows the policy improvement result in Lemma 5, (ii) follows the definition $\pi_{k+1}(a|s) := \dfrac{\exp\left(Q^{\text{soft}}_{r_{\theta_k},\pi_k}(s,a)\right)}{\sum_{\tilde{a}}\exp\left(Q^{\text{soft}}_{r_{\theta_k},\pi_k}(s,\tilde{a})\right)}$ in (5); (iii) follows the definition of the soft Bellman operator in (51).

For any $s\in\mathcal{S}$ and $a\in\mathcal{A}$, it holds that

$$0 \overset{(i)}{\leq} Q^{\text{soft}}_{r_{\theta_k},\pi_{\theta_k}}(s,a) - Q^{\text{soft}}_{r_{\theta_k},\pi_{k+1}}(s,a) \overset{(ii)}{\leq} Q^{\text{soft}}_{r_{\theta_k},\pi_{\theta_k}}(s,a) - \mathcal{T}_{\theta_k}(Q^{\text{soft}}_{r_{\theta_k},\pi_k})(s,a) \tag{53}$$

where (i) is due to the fact that $\pi_{\theta_k}$ is the optimal policy under reward parameter $\theta_k$; (ii) is from (52).

Hence, it further leads to

$$
\begin{aligned}
\|Q^{\text{soft}}_{r_{\theta_k},\pi_{\theta_k}} - Q^{\text{soft}}_{r_{\theta_k},\pi_{k+1}}\|_\infty &\overset{(i)}{\leq} \|Q^{\text{soft}}_{r_{\theta_k},\pi_{\theta_k}} - \mathcal{T}_{\theta_k}(Q^{\text{soft}}_{r_{\theta_k},\pi_k})\|_\infty\\
&\overset{(ii)}{=} \|\mathcal{T}_{\theta_k}(Q^{\text{soft}}_{r_{\theta_k},\pi_{\theta_k}}) - \mathcal{T}_{\theta_k}(Q^{\text{soft}}_{r_{\theta_k},\pi_k})\|_\infty\\
&\overset{(iii)}{\leq} \gamma\|Q^{\text{soft}}_{r_{\theta_k},\pi_{\theta_k}} - Q^{\text{soft}}_{r_{\theta_k},\pi_k}\|_\infty
\end{aligned}
\tag{54}
$$

where (i) is from (53); (ii) is from the fixed-point property in (17); (iii) is from the contraction property in (16). Therefore, we have the following result:

$$
\begin{aligned}
\|Q^{\text{soft}}_{r_{\theta_k},\pi_k} &- Q^{\text{soft}}_{r_{\theta_k},\pi_{\theta_k}}\|_\infty\\
&\overset{(i)}{\leq} \|Q^{\text{soft}}_{r_{\theta_{k-1}},\pi_k} - Q^{\text{soft}}_{r_{\theta_{k-1}},\pi_{\theta_{k-1}}}\|_\infty + 2L_q\|\theta_k - \theta_{k-1}\|\\
&\overset{(ii)}{\leq} \gamma\|Q^{\text{soft}}_{r_{\theta_{k-1}},\pi_{k-1}} - Q^{\text{soft}}_{r_{\theta_{k-1}},\pi_{\theta_{k-1}}}\|_\infty + 2L_q\|\theta_k - \theta_{k-1}\|
\end{aligned}
\tag{55}
$$

where (i) is from (50); (ii) is from (54).

To show the convergence of the soft Q-function based on (55), we further analyze the error between the reward parameters $\theta_k$ and $\theta_{k-1}$. Recall in Alg.1, the updates in reward parameters follows (8):

$$
\begin{aligned}
\theta_k &= \theta_{k-1} + \alpha g_{k-1}\\
&= \theta_{k-1} + \alpha\big(h(\theta_{k-1},\tau^E_{k-1}) - h(\theta_{k-1},\tau^A_{k-1})\big)
\end{aligned}
$$

where we denote $\tau = \{(s_t,a_t)\}_{t=0}^\infty$, $h(\theta,\tau) := \sum_{t\geq0}\gamma^t\nabla_\theta r(s_t,a_t;\theta)$ and $g_{k-1}$ is the stochastic gradient estimator at iteration $k-1$. Here, $\tau^E_{k-1}$ denotes the trajectory sampled from the expert's dataset $D$ at iteration $k-1$ and $\tau^A_{k-1}$ denotes the trajectory sampled from the agent's policy $\pi_k$ at time $k-1$. Then according to the inequality (9a) in Assumption 2, we could show that

$$\|g_{k-1}\| \leq \|h(\theta_{k-1},\tau^E_{k-1})\| + \|h(\theta_{k-1},\tau^A_{k-1})\| \leq 2L_r\sum_{t\geq0}\gamma^t = \frac{2L_r}{1-\gamma} = 2L_q \tag{56}$$

where the last equality follows the fact that we have defined the constant $L_q := \frac{L_r}{1-\gamma}$. Then we could further show that

$$
\begin{aligned}
&\|Q^{\text{soft}}_{r_{\theta_k},\pi_k} - Q^{\text{soft}}_{r_{\theta_k},\pi_{\theta_k}}\|_\infty \\
&\overset{(i)}{\leq} \gamma\|Q^{\text{soft}}_{r_{\theta_{k-1}},\pi_{k-1}} - Q^{\text{soft}}_{r_{\theta_{k-1}},\pi_{\theta_{k-1}}}\|_\infty + 2L_q\|\theta_k - \theta_{k-1}\| \\
&\overset{(ii)}{=} \gamma\|Q^{\text{soft}}_{r_{\theta_{k-1}},\pi_{k-1}} - Q^{\text{soft}}_{r_{\theta_{k-1}},\pi_{\theta_{k-1}}}\|_\infty + 2\alpha L_q\|g_{k-1}\| \\
&\overset{(iii)}{\leq} \gamma\|Q^{\text{soft}}_{r_{\theta_{k-1}},\pi_{k-1}} - Q^{\text{soft}}_{r_{\theta_{k-1}},\pi_{\theta_{k-1}}}\|_\infty + 4\alpha L_q^2
\end{aligned}
\tag{57}
$$

where (i) is from (55); (ii) follows the reward update scheme in (8); (iii) is from (56).

Summing the inequality (57) from $k = 1$ to $k = K$, it holds that

$$
\sum_{k=1}^{K} \|Q^{\text{soft}}_{r_{\theta_k},\pi_k} - Q^{\text{soft}}_{r_{\theta_k},\pi_{\theta_k}}\|_\infty \leq \gamma \sum_{k=0}^{K-1} \|Q^{\text{soft}}_{r_{\theta_k},\pi_k} - Q^{\text{soft}}_{r_{\theta_k},\pi_{\theta_k}}\|_\infty + 4\alpha K L_q^2
\tag{58}
$$

Rearranging the inequality (58) and divided (58) by $K$ on both sides, it holds that

$$
\frac{1-\gamma}{K} \sum_{k=1}^{K} \|Q^{\text{soft}}_{r_{\theta_k},\pi_k} - Q^{\text{soft}}_{r_{\theta_k},\pi_{\theta_k}}\|_\infty \leq \frac{\gamma}{K}\left(\|Q^{\text{soft}}_{r_{\theta_0},\pi_0} - Q^{\text{soft}}_{r_{\theta_0},\pi_{\theta_0}}\|_\infty - \|Q^{\text{soft}}_{r_{\theta_K},\pi_K} - Q^{\text{soft}}_{r_{\theta_K},\pi_{\theta_K}}\|_\infty\right) + 4\alpha L_q^2
\tag{59}
$$

Dividing the constant $1 - \gamma$ on both sides of (59), it holds that

$$
\frac{1}{K} \sum_{k=1}^{K} \|Q^{\text{soft}}_{r_{\theta_k},\pi_k} - Q^{\text{soft}}_{r_{\theta_k},\pi_{\theta_k}}\|_\infty \leq \frac{\gamma C_0}{K(1-\gamma)} + \frac{4L_q^2}{1-\gamma}\alpha
$$

where we denote $C_0 := \|Q^{\text{soft}}_{r_{\theta_0},\pi_0} - Q^{\text{soft}}_{r_{\theta_0},\pi_{\theta_0}}\|_\infty$. We could also write the inequality above as

$$
\begin{aligned}
&\frac{1}{K} \sum_{k=0}^{K-1} \|Q^{\text{soft}}_{r_{\theta_k},\pi_k} - Q^{\text{soft}}_{r_{\theta_k},\pi_{\theta_k}}\|_\infty \\
&\leq \frac{\gamma C_0}{T(1-\gamma)} + \frac{C_0}{T} - \frac{\|Q^{\text{soft}}_{r_{\theta_K},\pi_K} - Q^{\text{soft}}_{r_{\theta_K},\pi_{\theta_K}}\|_\infty}{K} + \frac{4L_q^2}{1-\gamma}\alpha \\
&\leq \frac{C_0}{T(1-\gamma)} + \frac{4L_q^2}{1-\gamma}\alpha.
\end{aligned}
$$

Recall the stepsize is defined as $\alpha = \frac{\alpha_0}{T^\sigma}$ where $\sigma > 0$. Then we have the following result:

$$
\frac{1}{K} \sum_{k=0}^{K-1} \|Q^{\text{soft}}_{r_{\theta_k},\pi_k} - Q^{\text{soft}}_{r_{\theta_k},\pi_{\theta_k}}\|_\infty = \mathcal{O}(K^{-1}) + \mathcal{O}(K^{-\sigma}).
\tag{60}
$$

With the inequality (49), it follows that

$$
\frac{1}{K} \sum_{k=0}^{K-1} \|\log \pi_{k+1} - \log \pi_{\theta_k}\|_\infty \leq \frac{2}{K} \sum_{k=0}^{K-1} \|Q^{\text{soft}}_{r_{\theta_k},\pi_k} - Q^{\text{soft}}_{r_{\theta_k},\pi_{\theta_k}}\|_\infty = \mathcal{O}(K^{-1}) + \mathcal{O}(K^{-\sigma}).
$$

Therefore, we complete the proof of (11a) in Theorem 2. $\qquad\square$

## H.2 Proof of (11b)

*Proof.* In this part, we prove the convergence of reward parameters $\{\theta_k\}_{k\geq 0}$.

We have the following result of the objective function $L(\theta)$:

$$
\begin{aligned}
L(\theta_{k+1}) &\overset{(i)}{\geq} L(\theta_k) + \langle \nabla L(\theta_k), \theta_{k+1} - \theta_k \rangle - \frac{L_c}{2}\|\theta_{k+1} - \theta_k\|^2 \\
&\overset{(ii)}{=} L(\theta_k) + \alpha \langle \nabla L(\theta_k), g_k \rangle - \frac{L_c \alpha^2}{2}\|g_k\|^2 \\
&= L(\theta_k) + \alpha \langle \nabla L(\theta_k), g_k - \nabla L(\theta_k) \rangle + \alpha\|\nabla L(\theta_k)\|^2 - \frac{L_c \alpha^2}{2}\|g_k\|^2 \\
&\overset{(iii)}{\geq} L(\theta_k) + \alpha \langle \nabla L(\theta_k), g_k - \nabla L(\theta_k) \rangle + \alpha\|\nabla L(\theta_k)\|^2 - 2L_c L_q^2 \alpha^2
\end{aligned}
\tag{61}
$$

where (i) is from the Lipschitz smooth property in (10b) of Lemma 2; (ii) follows the update scheme (8); (iii) is from constant bound in (56).

Taking an expectation over the both sides of (61), it holds that

$$
\mathbb{E}\left[L(\theta_{k+1})\right]
$$

$$
\geq \mathbb{E}\left[L(\theta_k)\right] + \alpha \mathbb{E}\left[\langle \nabla L(\theta_k), g_k - \nabla L(\theta_k) \rangle\right] + \alpha \mathbb{E}\left[\|\nabla L(\theta_k)\|^2\right] - 2L_c L_q^2 \alpha^2
$$

$$
= \mathbb{E}\left[L(\theta_k)\right] + \alpha \mathbb{E}\left[\langle \nabla L(\theta_k), \mathbb{E}\left[g_k - \nabla L(\theta_k)\big|\theta_k\right]\rangle\right] + \alpha \mathbb{E}\left[\|\nabla L(\theta_k)\|^2\right] - 2L_c L_q^2 \alpha^2
$$

$$
\overset{(i)}{=} \mathbb{E}\left[L(\theta_k)\right] + \alpha \mathbb{E}\left[\left\langle \nabla L(\theta_k), \mathbb{E}_{\tau \sim \pi_{\theta_k}}\left[\sum_{t \geq 0} \gamma^t \nabla_\theta r(s_t, a_t; \theta_t)\right] - \mathbb{E}_{\tau \sim \pi_{k+1}}\left[\sum_{t \geq 0} \gamma^t \nabla_\theta r(s_t, a_t; \theta_t)\right]\right\rangle\right]
$$

$$
+ \alpha \mathbb{E}\left[\|\nabla L(\theta_k)\|^2\right] - 2L_c L_q^2 \alpha^2
$$

$$
\overset{(ii)}{\geq} \mathbb{E}\left[L(\theta_k)\right] - 2\alpha L_q \underbrace{\mathbb{E}\left[\left\|\mathbb{E}_{\tau \sim \pi_{\theta_k}}\left[\sum_{t \geq 0} \gamma^t \nabla_\theta r(s_t, a_t; \theta_k)\right] - \mathbb{E}_{\tau \sim \pi_{k+1}}\left[\sum_{t \geq 0} \gamma^t \nabla_\theta r(s_t, a_t; \theta_k)\right]\right\|\right]}_{\text{term A}}
$$

$$
+ \alpha \mathbb{E}\left[\|\nabla L(\theta_k)\|^2\right] - 2L_c L_q^2 \alpha^2
\tag{62}
$$

where (i) follows (6) and (7); (ii) is due to the fact that $\|\nabla L(\theta)\| \leq 2L_q$.

Then we further analyze the term A as below:

$$
\mathbb{E}\left[\left\|\mathbb{E}_{\tau \sim \pi_{\theta_k}}\left[\sum_{t \geq 0} \gamma^t \nabla_\theta r(s_t, a_t; \theta_k)\right] - \mathbb{E}_{\tau \sim \pi_{k+1}}\left[\sum_{t \geq 0} \gamma^t \nabla_\theta r(s_t, a_t; \theta_k)\right]\right\|\right]
$$

$$
\overset{(i)}{=} \mathbb{E}\left[\left\|\frac{1}{1-\gamma}\mathbb{E}_{(s,a) \sim d(\cdot,\cdot;\pi_{\theta_k})}\left[\nabla_\theta r(s, a; \theta_k)\right] - \frac{1}{1-\gamma}\mathbb{E}_{(s,a) \sim d(\cdot,\cdot;\pi_{k+1})}\left[\nabla_\theta r(s, a; \theta_k)\right]\right\|\right]
$$

$$
\overset{(ii)}{\leq} \frac{2}{1-\gamma} \cdot \max_{s \in \mathcal{S}, a \in \mathcal{A}} \|\nabla_\theta r(s, a; \theta_k)\| \cdot \mathbb{E}\left[\|d(\cdot,\cdot;\pi_{\theta_k}) - d(\cdot,\cdot;\pi_{k+1})\|_{TV}\right]
$$

$$
\overset{(iii)}{\leq} \frac{2L_r}{1-\gamma}\mathbb{E}\left[\|d(\cdot,\cdot;\pi_{\theta_k}) - d(\cdot,\cdot;\pi_{k+1})\|_{TV}\right]
$$

$$
\overset{(iv)}{\leq} 2L_q C_d \mathbb{E}\left[\|Q^{\text{soft}}_{r_{\theta_k},\pi_{\theta_k}} - Q^{\text{soft}}_{r_{\theta_k},\pi_k}\|\right]
$$

$$
\overset{(v)}{\leq} 2L_q C_d \sqrt{|\mathcal{S}| \cdot |\mathcal{A}|}\, \mathbb{E}\left[\|Q^{\text{soft}}_{r_{\theta_k},\pi_{\theta_k}} - Q^{\text{soft}}_{r_{\theta_k},\pi_k}\|_\infty\right]
\tag{63}
$$

where (i) follows the definition $d(s,a;\pi) = (1-\gamma)\pi(a|s)\sum_{t \geq 0}\gamma^t \mathcal{P}^\pi(s_t = s|s_0 \sim \eta)$; (ii) is due to distribution mismatch between two visitation measures; (iii) follows the inequality (9a) in Assumption 2; the inequality (iv) follows Lemma 4 and the fact that $\pi_{\theta_k}(\cdot|s) \propto \exp\left(Q^{\text{soft}}_{r_{\theta_k},\pi_{\theta_k}}(s,\cdot)\right)$, $\pi_{k+1}(\cdot|s) \propto \exp\left(Q^{\text{soft}}_{r_{\theta_k},\pi_k}(s,\cdot)\right)$ and the constant $L_q := \frac{L_r}{1-\gamma}$; (v) follows the conversion between

Frobenius norm and infinity norm. Through plugging the inequality (63) into (62), it leads to

$$\mathbb{E}\left[L(\theta_{k+1})\right]$$

$$\geq \mathbb{E}\left[L(\theta_k)\right] - 2\alpha L_q \mathbb{E}\left[\left\|\mathbb{E}_{\tau \sim \pi_{\theta_k}}\left[\sum_{t \geq 0} \gamma^t \nabla_\theta r(s_t, a_t; \theta_k)\right] - \mathbb{E}_{\tau \sim \pi_{k+1}}\left[\sum_{t \geq 0} \gamma^t \nabla_\theta r(s_t, a_t; \theta_k)\right]\right\|\right]$$

$$+ \alpha \mathbb{E}\left[\|\nabla L(\theta_k)\|^2\right] - 2L_c L_q^2 \alpha^2$$

$$\overset{(i)}{\geq} \mathbb{E}\left[L(\theta_k)\right] - 4\alpha C_d L_q^2 \sqrt{|\mathcal{S}| \cdot |\mathcal{A}|} \mathbb{E}\left[\|Q^{\text{soft}}_{r_{\theta_k}, \pi_{\theta_k}} - Q^{\text{soft}}_{r_{\theta_k}, \pi_k}\|_\infty\right] + \alpha \mathbb{E}\left[\|\nabla L(\theta_k)\|^2\right] - 2L_c L_q^2 \alpha^2$$

where (i) follows the inequality (63).

Rearranging the inequality above and denote $C_1 := 4C_d L_q^2 \sqrt{|\mathcal{S}| \cdot |\mathcal{A}|}$, it holds that

$$\alpha \mathbb{E}\left[\|\nabla L(\theta_k)\|^2\right] \leq 2L_c L_q^2 \alpha^2 + \alpha C_1 \mathbb{E}\left[\|Q^{\text{soft}}_{r_{\theta_k}, \pi_{\theta_k}} - Q^{\text{soft}}_{r_{\theta_k}, \pi_k}\|_\infty\right] + \mathbb{E}\left[L(\theta_{k+1}) - L(\theta_k)\right]$$

Summing the inequality above from $k = 0$ to $K - 1$ and dividing both sides by $\alpha K$, it holds that

$$\frac{1}{K}\sum_{k=0}^{K-1} \mathbb{E}\left[\|\nabla L(\theta_k)\|^2\right] \leq 2L_c L_q^2 \alpha + \frac{C_1}{K}\sum_{k=0}^{K-1} \mathbb{E}\left[\|Q^{\text{soft}}_{r_{\theta_k}, \pi_{\theta_k}} - Q^{\text{soft}}_{r_{\theta_k}, \pi_k}\|_\infty\right] + \mathbb{E}\left[\frac{L(\theta_K) - L(\theta_0)}{K\alpha}\right]$$
$$\tag{64}$$

Note that the log-likelihood function $L(\theta_K)$ is negative and $L(\theta_0)$ is a bounded constant. Then we could plug (60) into (64), it holds that

$$\frac{1}{K}\sum_{K=0}^{K-1} \mathbb{E}\left[\|\nabla L(\theta_K)\|^2\right] = \mathcal{O}(K^{-\sigma}) + \mathcal{O}(K^{-1}) + \mathcal{O}(K^{-1+\sigma}) \tag{65}$$

which completes the proof for the inequality (11b). $\qquad\square$