# OpenReview forum: "Maximum-Likelihood Inverse Reinforcement Learning with Finite-Time Guarantees"
_NeurIPS.cc/2022/Conference — NeurIPS 2022 Accept_

### Official Review · Reviewer_wFjp · 2022-07-05

**Rating:** 5
**Confidence:** 4
**Soundness:** 3 good
**Presentation:** 2 fair
**Contribution:** 3 good

**Summary:**

In this paper, the authors proposed a new formulation of IRL based on maximum likelihood, which is equivalent to MaxEnt when the reward function is linear.  By leveraging this formulation, the authors propose a new computationally efficient gradient-based iterative algorithm that does not require solving MDP in each iteration. The authors further prove a non-asymptotically rate of how fast the algorithm converges to a stationary point, which is the first non-asymptotic convergence result for ILR with nonlinear reward parameterization. Finally, the authors conduct extensive experiments that show that the proposed algorithm outperforms several state-of-the-art IRL algorithms.

**Questions:**

In the algorithm, it’s assumed that the soft-Q function can be accurately estimated. Can the approximation  in soft-Q function be considered in the theorem as well?

Interestingly, the gradient in Eq.(6) is the same as the gradient to r with fixed pi in Eq. (2) (MaxEnt). And the inner optimization in Eq. (2) is finding the optimal policy given r which is the same as solving Eq. (3a). Therefore, the proposed algorithm can be considered as solving Eq. (2) directly without the ML-IRL formulation. Would the authors provide some explanation on why formulating IRL in ML-IRL is fundamental to the convergence proof?

In the text the authors try to distinguish between MaxEnt IRL and adversarial IRL (e.g. Line 30 - 44). I wonder why that's the case as these two have been shown to be closely related [6, 7].

[6] Finn, Chelsea, et al. "A connection between generative adversarial networks, inverse reinforcement learning, and energy-based models." arXiv preprint arXiv:1611.03852 (2016).

[7] Swamy, Gokul, et al. "Of moments and matching: A game-theoretic framework for closing the imitation gap." International Conference on Machine Learning. PMLR, 2021.


**Limitations:**

Yes

**Strengths And Weaknesses:**

### Strength:

The authors claim that Thm 2 is the first non-asymptotic convergence result for ILR with nonlinear reward parameterization. I believe this is true as a recent paper [1] from ICML 2021 proves a non-asymptotic rate for a gradient-based method with linear reward parameterization (although that paper’s proof does not seem to have the assumption the value function can be accurately estimated).

### Weakness:
 I think the contribution above is important to the IRL community.  However, other contributions of this paper may not be as significant as this one.

1. The new formulation of IRL. This maximum likelihood formulation is known [2]. The proof seems to be similar to Thm 3 in [3].
2. The algorithm. The authors mentioned that the algorithm enjoys computational efficiency and is capable of reward transferring. However, GAIL-type of algorithms that use state-dependent rewards in the discriminator [4] and scalable MaxEnt IRL algorithms [5] are able to achieve those benefits as well.
3. The experiments. I like the experiments on reward transferring. However, the main goal of proposing the new algorithm and theory is to reduce computation burden, but the experiments focus on the quality of the learned policy / reward after convergence. It’s unclear from the experiments whether the new algorithm is indeed faster. For example, an interesting experiment would be a comparison with MaxEnt IRL as in theory the proposed method is able to converge to a stationary point that’s the same as MaxEnt IRL, but faster.

Furthermore, the clarity of the paper can be improved. Line 118 to 135 is rather confusing and only until the second reading I was able to understand the math. In Line 119,\pi_\theta is parameterized by \theta. But in Line 123, \theta is used to parameterize the reward function.

[1] Kamoutsi, Angeliki, Goran Banjac, and John Lygeros. "Efficient Performance Bounds for Primal-Dual Reinforcement Learning from Demonstrations." International Conference on Machine Learning. PMLR, 2021.

[2] Jain, Vinamra, Prashant Doshi, and Bikramjit Banerjee. "Model-free IRL using maximum likelihood estimation." Proceedings of the AAAI Conference on Artificial Intelligence. Vol. 33. No. 01. 2019.

[3] Ziebart, Brian D., J. Andrew Bagnell, and Anind K. Dey. "The principle of maximum causal entropy for estimating interacting processes." IEEE Transactions on Information Theory 59.4 (2013): 1966-1980.

[4] Torabi, Faraz, Garrett Warnell, and Peter Stone. "Generative adversarial imitation from observation." arXiv preprint arXiv:1807.06158 (2018).

[5] Finn, Chelsea, Sergey Levine, and Pieter Abbeel. "Guided cost learning: Deep inverse optimal control via policy optimization." International conference on machine learning. PMLR, 2016.

---

> ### Author Response · Authors · 2022-08-02
> **Author's Response: Contributions and Clarifications**
>
> We thank the reviewer for the detailed review of the paper and the valuable feedback. Below, we address the reviewer's comments in a point-by-point manner.
>
> **Weakness 1**: The new formulation of IRL. This maximum likelihood formulation is known [Jain, et al, 2019]. The proof seems to be similar to Thm 3 in [Ziebart, et al, 2013].
>
> **Our Response**: We believe there are major differences.
>
> * [**The formulation**]. As we mentioned in our paper, [Jain, et al, 2019] also consider a maximum likelihood formulation. However, the formulation of  [Jain, et al, 2019] only stands under **finite horizon** and the policy is not clearly identified in [Jain, et al, 2019]. In contrast, we consider the objective as a likelihood function under the **infinite horizon**, and rigorously identified the policy as the optimal solution of an entropy-regularized MDP. Furthermore, we develop a duality theory to show the correctness and the reasonability of our problem formulation ML-IRL.
>
> * [**The proof**]. In Theorem 3 in [Ziebart, et al, 2013], the setting is on an interacting process with a finite horizon.
> Compared with Theorem 3 in [Ziebart, et al, 2013], there are three major differences in our proof. First, we consider the Markov decision process so that the Markov property should be utilized to develop more specific results. Second, we are considering the infinite horizon. Third, in our duality analysis, we rigorously identify that the policy in ML-IRL is the solution of an entropy-regularized MDP.
>
> In the end, we would like to emphasize the main contribution of this paper does not reside in the problem formulation.
> The algorithm proposed in this paper is the **first provable IRL algorithm with finite-time analysis under nonlinear reward parameterization**.
>
>
> **Weakness 2**: The algorithm. The authors mentioned that the algorithm enjoys computational efficiency and is capable of reward transferring. However, GAIL-type of algorithms that use state-dependent rewards in the discriminator [Torabi, et al, 2018] and scalable MaxEnt IRL algorithms [Finn, et al, 2016] are able to achieve those benefits as well.
>
> **Our Response**: Thanks for this good question. Let’s clarify the difference from two perspectives:
>
> * First we want to emphasize that most GAIL-type algorithms are imitation learning algorithms, which enjoy computational efficiency from the single-loop training but the resulting discriminator could not be reused as a stationary reward function and transferred to new environments [1]. Moreover, [2,3] pointed out that the brittle approximation techniques and sensitive hyperparameter choice in GAIL-type algorithms could induce the instability in the training process.
>
> * Guided cost learning (GCL) samples the trajectory from the current estimate of the trajectory distribution and the sampled trajectories are used to estimate the gradient. This is a form of importance sampling. The computational burden of sampling paths with long planning horizons and high-dimensional state space is significant. Hence, implementations of GCL use ad-hoc approximation techniques that often induce instabilities. It must be emphasized that GCL has no theoretical guarantees for finite-time performance. In the proposed algorithm ML-IRL, the inner loop is replaced with a policy update for the entropy-regularized MDP defined by the current reward parameter estimate. Moreover, ML-IRL samples trajectories from the updated policy for rewardt gradient estimation.  Note that sampling from a Markovian policy is much simpler than sampling from a trajectory distribution as in GCL. In addition, we could easily use state-of-the-art methods like soft Q-learning or soft Actor-Critic to implement the policy update. In contrast to GCL, our proposed algorithm has a finite-time performance guarantee.
>
> [1] Ni, Tianwei, et al. "f-irl: Inverse reinforcement learning via state marginal matching." arXiv preprint arXiv:2011.04709 (2020).
>
> [2] Kurach, Karol, et al. "The gan landscape: Losses, architectures, regularization, and normalization." (2018).
>
> [3] Kostrikov, Ilya, et al. "Discriminator-actor-critic: Addressing sample inefficiency and reward bias in adversarial imitation learning." arXiv preprint arXiv:1809.02925 (2018).

---

> > ### Author Response · Authors · 2022-08-02
> > **Response Continued: Contributions and Clarifications**
> >
> > **Weakness 3**: The experiments. I like the experiments on reward transferring. However, the main goal of proposing the new algorithm and theory is to reduce computation burden, but the experiments focus on the quality of the learned policy / reward after convergence. It’s unclear from the experiments whether the new algorithm is indeed faster. For example, an interesting experiment would be a comparison with MaxEnt IRL as in theory the proposed method is able to converge to a stationary point that’s the same as MaxEnt IRL, but faster.
> >
> > **Our Response**: Thanks for the comments! A new experiment is added in Appendix B.3.
> >
> > According to the figures of reward visualization in Appendix B.3, we could observe that ML-IRL and MaxEnt-IRL have similar estimation accuracy on the recovered rewards. Moreover, from the convergence curves about the likelihood values, we could see that ML-IRL is much more efficient than MaxEnt-IRL in terms of the soft policy iteration number.
> >
> > **Weakness 4**: Furthermore, the clarity of the paper can be improved. Line 118 to 135 is rather confusing and only until the second reading I was able to understand the math. In Line 119, $\pi_\theta$ is parameterized by $\theta$. But in Line 123, $\theta$ is used to parameterize the reward function.
> >
> > **Our Response**: Thanks for the comments. We have revised the description from line 118 to 135.
> >
> > We clarify that $\theta$ is the reward parameter. Moreover, we define $\pi_{\theta}$ as the solution to an entropy-regularized MDP, when the underlying reward function is $r(\cdot, \cdot; \theta)$.
> >
> > **Question 1**: In the algorithm, it’s assumed that the soft-Q function can be accurately estimated. Can the approximation in soft-Q function be considered in the theorem as well?
> >
> > **Our Response**: Thanks for this interesting question! We believe the approximation in soft Q-function could be analyzed as well.
> >
> > Suppose that at a given iteration $k$ the soft Q-function ${Q}\_{k}$ is approximated by $ \widehat{Q}\_{k}$, the new policy $\pi_{k+1}$ could be generated by an **approximated soft policy iteration**: $\pi_{k+1}(a|s) \propto \exp( \widehat{Q}\_{k}(s,a) )$ for any state-action pair (s,a). Then our algorithm can still be executed.
> >
> > In the analysis, given an approximation error $\epsilon$, if $||\widehat{Q}\_{k} - {Q}\_{k}||\_{\infty} \leq \epsilon$ for each iteration $k$, then we expect that we can still obtain similar theoretical results. **In this case, we expect that the final convergence rate will be $\mathcal{O}(K^{-1/2}) +  \mathcal{O}(\epsilon)$ for both reward parameter and policy.** That is, the approximation error $\epsilon$ will appear in the final rate as a constant.
> >
> > However, new proof techniques are required to analyze the convergence of the “approximated” soft policy iteration. Considering such analysis will be nontrivial and there is no space remaining in our paper, we expect to consider this extension of our analysis as a future work.

---

> > > ### Author Response · Authors · 2022-08-02
> > > **Response Continued: Contributions and Clarifications**
> > >
> > > **Question 2**: Interestingly, the gradient in Eq.(6) is the same as the gradient to r with fixed pi in Eq. (2) (MaxEnt). And the inner optimization in Eq. (2) is finding the optimal policy given r which is the same as solving Eq. (3a). Therefore, the proposed algorithm can be considered as solving Eq. (2) directly without the ML-IRL formulation. Would the authors provide some explanation on why formulating IRL in ML-IRL is fundamental to the convergence proof?
> > >
> > > **Our Response**: Thanks for this interesting question. It is true that our algorithm also solves (2). However, (2) is not MaxEnt-IRL.
> > >
> > > Only under linear reward parameterization, (2) is equivalent to (1) which is the Lagrangian function of MaxEnt-IRL. However, when the reward function is nonlinear, (2) is not related to MaxEntIRL and loses its practical meaning. Therefore, we need to consider a new formulation to understand IRL under nonlinear reward parameterization.
> > >
> > > We propose ML-IRL from the maximum likelihood perspective, and then formally identify it as the dual problem of MaxEnt-IRL under linear reward parameterization. After that, we further claim ML-IRL is a better formulation than MaxEnt-IRL, since ML-IRL has good reasonability and is general enough to include nonlinear reward parameterization.
> > >
> > > Moreover, we expect the structure of the bi-level optimization problem in ML-IRL could help readers better understand the motivations behind two-timescale / singe-loop approaches.
> > >
> > >
> > > **Question 3**: In the text the authors try to distinguish between MaxEnt IRL and adversarial IRL (e.g. Line 30 - 44). I wonder why that's the case as these two have been shown to be closely related [Finn, et al, 2016] and [Swamy, et al, 2021].
> > >
> > > **Our Response**: Thanks for this question. First, we admit that these two are closely related. However, we think MaxEnt-IRL is a classic formulation, while adversarial IRL are recently proposed algorithms which leverage the adversarial training in GAN to solve the MaxEnt-IRL problem.

---

> > > > ### Comment · Reviewer_wFjp · 2022-08-09
> > > > **Thanks for the reply!**
> > > >
> > > > I really appreciate the detailed response from the authors. It’s very helpful. I increased my score. However, I still have concerns on experimental results (the added one is on grid world) and impact in general (because the algorithm does not seem novel).

---

> > > > > ### Author Response · Authors · 2022-08-09
> > > > > **Many thanks for your comments and evaluations**
> > > > >
> > > > > Thank you very much! We sincerely appreciate your comments and evaluations, which also help us to improve our paper.
> > > > >
> > > > > We believe our follow-up discussion with Reviewer fBA5 is directly related to your concern pertaining to novelty and impact. We acknowledge that the two-time scale approximation algorithm framework has already been used in related work, like GAIL [1] and GCL [2]. However, there are major differences between GAIL / GCL and ML-IRL. Compared with ML-IRL, GAIL has the difficulty to recover the ground truth reward function. Moreover, GAIL-type algorithms also suffer from the instability in the adversarial training. Furthermore, in contrast to ML-IRL which models the agent’s policy as a solution to a maximum entropy RL problem, GCL directly models the trajectory distribution as a solution to an entropy regularized trajectory problem. Such modeling in GCL could induce unstable training especially when the time horizon is large or the dimension is high. (We also present a detailed discussion to show the difference between ML-IRL and GCL. It is in our follow-up discussion with Reviewer fBA5: https://openreview.net/forum?id=zbt3VmTsRIj&noteId=OfhbvX5XGlt)
> > > > >
> > > > > Considering we provide the first finite-time convergence results in IRL with nonlinear reward parameterization, we hope this contribution could also address your concern pertaining to novelty and impact.
> > > > >
> > > > > Thank you again for taking the time to review our paper!

---

> ### Author Response · Authors · 2022-08-07
> **Looking Forward to Post-Rebuttal Feedback**
>
> Dear reviewer wFjp,
>
> Thank you very much for taking the time to review our paper! We cherish your comments and evaluations very much! In our posted responses, we have made a point-to-point response to alleviate your concerns. If you have any further concerns on our response, we are more than happy to address them.
>
> Best,
>
> Authors

---

### Official Review · Reviewer_pJ4m · 2022-07-09

**Rating:** 5
**Confidence:** 3
**Soundness:** 3 good
**Presentation:** 3 good
**Contribution:** 3 good

**Summary:**

This paper proposes a novel single-loop IRL algorithm to reduce the computational burden of the previous IRL framework. The authors prove that the solution of the proposed framework is equivalent to the maximum entropy IRL framework with a  linearly parameterized reward function. They also provide a convergence guarantee of the proposed algorithm. The experiment result seems promising.

**Questions:**

- Could the authors discuss the MaxEnt-IRL literature?
- Could the authors explain why the ML-IRL framework works for the transfer learning tasks?

**Limitations:**

The authors adequately addressed the limitations and potential negative societal impact of their work.

**Strengths And Weaknesses:**

Strengths:

- The paper is well written.
- The problem studied in this paper is important and the proposed method is novel.
- The authors provide theoretical guarantees for the proposed framework.

Weaknesses:

- There may need more literature review on the MaxEnt-IRL.
- Although the ML-IRL framework could reduce the computational complexity, it is not obvious why the proposed ML-IRL framework works for the transfer learning tasks.

---

> ### Author Response · Authors · 2022-08-02
> **Response: Contributions and Clarifications**
>
> We thank the reviewer for the detailed review of the paper and the valuable feedback. Below, we address the reviewer's comments in a point-by-point manner.
>
> **Your Comments (Weakness 1 & Question 1)**: There may need more literature review on the MaxEnt-IRL. Could the authors discuss the MaxEnt-IRL literature?
>
> **Our Response**: Thanks for the comments. In our original version, most related work about MaxEnt-IRL is discussed on the introduction section and Appendix A. In our revised version, we include more related work and present a detailed discussion around MaxEnt-IRL in Appendix A.
>
> **Your Comments (Weakness 2 & Question 2)**: Although the ML-IRL framework could reduce the computational complexity, it is not obvious why the proposed ML-IRL framework works for the transfer learning tasks. Could the authors explain why the ML-IRL framework works for the transfer learning tasks?
>
> **Our Response**: Thanks for this good question. We would like to explain this from two perspectives:
>
> * In the related work of inverse reinforcement learning, [1,2] pointed out that state-only reward function is more robust under the transfer learning settings. Intuitively, the demonstrated actions from the expert will not provide any useful information in the transfer learning setting, since the environment dynamic is different and the same action will not lead to the same transition under different environment dynamics. Therefore, we could parameterize the reward as a function only dependent on the state, so that ML-IRL could potentially avoid overfitting the demonstrated actions from the expert.
>
> * Moreover, there are theoretical results [3,4] which show that MaxEnt-IRL is able to identify transferable reward functions. Since the ML-IRL framework is close to MaxEnt-IRL (equivalent under linear reward parameterization), it is reasonable to expect that ML-IRL could also work well for the transfer learning tasks.
>
> [1] Ni, Tianwei, et al. "f-irl: Inverse reinforcement learning via state marginal matching." arXiv preprint arXiv:2011.04709 (2020).
>
> [2] Viano, Luca, et al. "Robust inverse reinforcement learning under transition dynamics mismatch." Advances in Neural Information Processing Systems 34 (2021): 25917-25931.
>
> [3] Kim, Kuno, et al. "Reward identification in inverse reinforcement learning." International Conference on Machine Learning. PMLR, 2021.
>
> [4] Cao, Haoyang, Samuel Cohen, and Lukasz Szpruch. "Identifiability in inverse reinforcement learning." Advances in Neural Information Processing Systems 34 (2021): 12362-12373.

---

> ### Author Response · Authors · 2022-08-07
> **Looking Forward to Post-Rebuttal Feedback**
>
> Dear reviewer pJ4m,
>
> Thank you very much for taking the time to review our paper! We cherish your comments and evaluations very much! In our posted responses, we have made a point-to-point response to alleviate your concerns. If you have any further concerns on our response, we are more than happy to address them.
>
> Best,
>
> Authors

---

### Official Review · Reviewer_6Mga · 2022-07-10

**Rating:** 6
**Confidence:** 3
**Soundness:** 3 good
**Presentation:** 2 fair
**Contribution:** 3 good

**Summary:**

The paper formalizes IRL as a entropy-regularized maximum likelihood problem. The authors show that this problem is dual to maximum entropy IRL, and then formulates this problem as a bi-level optimization. They propose an algorithm that iterates between single steps of evaluation and policy improvement (in contrast to most max entropy IRL methods that have to solve RL problems in an inner loop). For this algorithm, the authors provide a _finite time_ convergence guarantee under some regularity conditions. Finally, they provide an empirical evaluation on standard MuJoCo task, demonstrating that their algorithm outperforms alternative IRL approaches.

**Questions:**

Can you provide standard errors for the results in Table 1?

Why does the state-action version of the algorithm perform worse than the state-only algorithm in the half-cheetah environment?


**Typos:**
 - "inherent nested" -> "inherently nested" (line 4)
- "Mujoco" -> "MuJoCo" (line 19)
- "computational efficient" -> "computationally efficient" (line 53)

**Limitations:**

I did not find any discussion of potential broader impact of this work.

**Strengths And Weaknesses:**

**Strengths**
- The paper provides the first finite-time convergence analysis without assuming linear rewards, as far as I can tell. The assumptions of the main theoretical results seem pretty reasonable; this is a solid theoretical contribution.
- The duality between MaxEnt IRL and the proposed maximum likelihood formulation might be interesting to the IRL community and could motivate interesting follow-up work.
- The authors provide a good experimental evaluation that covers most relevant baselines and shows that their algorithm can perform well empirically.
- The authors discuss the novelty of their work and provide particularly appropriate comparisons to results in prior work at multiple point, e.g., for their ML formulation, the proposed algorithm, and the main theoretical result.


**Weaknesses**
- The duality with MaxEnt IRL is not particularly novel and has been observed before in other context (as noted by the authors).
- The results in Table 1 don't show very large differences, and it is hard to evaluate if these are significant differences, given that only 3 random seeds were run.
- The experimental evaluation is limited to MuJoCo locomotion tasks which can sometimes be too simple to draw strong conclusions.
- Overall, the writing could be clearer at times, and the structure of the paper could be improved. For example, the Introduction spends a lot of time discussion prior work before stating the contributions of this paper, which is sub-optimal.

---

> ### Author Response · Authors · 2022-08-02
> **Author's Response**
>
> We appreciate your review and the positive comments regarding our paper. We would like to respond to your comments and hope that we address all your concerns below:
>
> **Weakness 1**: The duality with MaxEnt IRL is not particularly novel and has been observed before in other context (as noted by the authors).
>
> **Our Response**: To our knowledge there is no formal proof of such duality in the literature. The reference [1] proves a similar duality result for a **finite horizon** interacting process. Our proof is for the **infinite horizon** MDP, which is not a straightforward extension of the duality result in [1]. We agree that generally speaking the duality between maximum entropy and maximum likelihood is well-known. However, for the specific problems we consider (maximum likelihood IRL problem characterized by infinite horizon MDP), proving duality requires additional effort in order to carefully identify the precise duality pairs.
>
>
> **Weakness 2**: The results in Table 1 don't show very large differences, and it is hard to evaluate if these are significant differences, given that only 3 random seeds were run.
>
> **Our Response**: Thanks for this comment. We have added up to 6 random seeds and Table 1 in the revised version is updated. Due to the limited space, we present the numerical results with confidence intervals in Appendix (see Table 3).
>
> **Weakness 3**: The experimental evaluation is limited to MuJoCo locomotion tasks which can sometimes be too simple to draw strong conclusions.
>
> **Our Response**: We observed that MuJoCo is a common platform for algorithm evaluation, and the state-of-the-art imitation learning / IRL algorithms [1,2,3] are also evaluated in MuJoCo. Since we would like to compare our algorithms with SOTA,  therefore we think MoJoCo is the best platform to use.
>
> **Weakness 4**: Overall, the writing could be clearer at times, and the structure of the paper could be improved. For example, the Introduction spends a lot of time discussion prior work before stating the contributions of this paper, which is sub-optimal.
>
> **Our Response**: Thanks for the comments. We have revised the paper and highlighted the changes in blue.
>
> **Question 1**: Can you provide standard errors for the results in Table 1?
>
> **Our Response**: Yes, we did. Now, we add up to 6 random seeds. Due to the limited space of the main paper, we present the numerical results with confidence intervals in the Appendix (see Table 3).
>
> **Question 2**: Why does the state-action version of the algorithm perform worse than the state-only algorithm in the half-cheetah environment?
>
> **Our Response**: Among the MuJoCo benchmark tasks, it is reasonable that ML-IRL (state-only reward) may have better performance than ML-IRL (state-action reward) in some tasks.
>
> For standard IRL tasks, we observe that ML-IRL (state-only reward) and ML-IRL (state-action reward) perform similarly. The numerical results are in Table 1 and Table 3. Due to statistical variance of the sampling procedure (see the confidence intervals in Table 3), it is possible that for some tasks one algorithm is slightly better than the other algorithm, while for other tasks it is reversed.
>
> **Limitations**: I did not find any discussion of potential broader impact of this work.
>
> **Our Response**: Thanks for the comments. A paragraph on potential broader impacts has been added to the revision version (see the last paragraph in Appendix A).
>
> In our original version, we have already discussed the limitations of our work in the conclusion. Moreover, we have also mentioned the potential negative social impacts at the beginning of the Appendix.
>
> [1] Ho, Jonathan, and Stefano Ermon. "Generative adversarial imitation learning." Advances in neural information processing systems 29 (2016).
>
> [2] Ni, Tianwei, et al. "f-irl: Inverse reinforcement learning via state marginal matching." arXiv preprint arXiv:2011.04709 (2020).
>
> [3] Garg, Divyansh, et al. "IQ-Learn: Inverse soft-Q Learning for Imitation." Advances in Neural Information Processing Systems 34 (2021): 4028-4039.

---

> > ### Comment · Reviewer_6Mga · 2022-08-06
> > **Thanks for the response!**
> >
> > I appreciate the authors' response and clarifications! I still think the paper should be accepted, but my primary concerns about novelty and empirical evaluation remain. So, I still agree with my original evaluation and I will keep my score as "Weak Accept".

---

> > > ### Author Response · Authors · 2022-08-09
> > > **Many thanks for your comments and positive evaluations**
> > >
> > > We sincerely appreciate you for taking time to review our paper and thank you for recognizing the contributions of this work.

---

### Official Review · Reviewer_fBA5 · 2022-07-12

**Rating:** 6
**Confidence:** 3
**Soundness:** 3 good
**Presentation:** 2 fair
**Contribution:** 3 good

**Summary:**

The authors propose a maximum likelihood approach for IRL. Theoretical results show that the algorithm converges in finite time and that for linear rewards that the ML-IRL problem is dual with MaxEnt-IRL and that there is strong duality. Empirical results show that ML-IRL outperforms state-of-the-art IRL algorithms across several standard benchmarks.

**Questions:**

The authors are missing related work that also seeks to make IRL more tractable.

Wang, Ruohan, et al. "Random expert distillation: Imitation learning via expert policy support estimation." International Conference on Machine Learning. PMLR, 2019.
and
Brown, Daniel S., Wonjoon Goo, and Scott Niekum. "Better-than-demonstrator imitation learning via automatically-ranked demonstrations." Conference on robot learning. PMLR, 2020.
Propose approaches that do not require interleaving reward learning and policy learning.
Recent work
Barde, Paul, et al. "Adversarial soft advantage fitting: Imitation learning without policy optimization." Advances in Neural Information Processing Systems 33 (2020): 12334-12344.
proposes an adversarial IL approach that does not require any RL.

One of the stated contributions is to implement  single-loop approach. This doesn't seem like a contribution since most Adversarial IRL methods do the same thing. This idea is well studied, going back to GAIL and Guided cost learning, and probably earlier. I do not think the authors can claim this as novel. Furthermore, the two-timescale approach is common in GANs and GAIL-like algorithms. I don't think this can be claimed as a contribution.

Equation 3: The theta parameterizes r which informs argmax to get pi. This is confusing with line 119 where the policy is parameterized by theta. Policies are not usually parameterized by rewards.

Line 155: is supposed to be an example of solving a hard optimization problem in the inner loop but it refers to IQ-Learn which does was previously noted to be computationally efficient because it avoids this inner loop mdp solver.

Lemma 1: How is his different from the guided cost learning update. In guided cost learning the partition function is estimated with onpolicy rollouts so it appears quite similar.

Line 188: Why only sample one demo and one policy rollout? Doesn't this have very high variance?

Allowing the algorithm to work with state only demonstrations is nice, but many other algorithms already do this. E.g.
Brown, Daniel S., Wonjoon Goo, and Scott Niekum. "Better-than-demonstrator imitation learning via automatically-ranked demonstrations." Conference on robot learning. PMLR, 2020.
Torabi, Faraz, Garrett Warnell, and Peter Stone. "Generative adversarial imitation from observation." arXiv preprint arXiv:1807.06158 (2018).
As well as all of the classical IRL algorithms.
It would be good to motivate more why this is a contribution.


**Limitations:**

A discussion of limitations are lacking in the main body of the paper.

**Strengths And Weaknesses:**

+The strong duality theorem appears novel and draws an interesting connection between ML-IRL and MaxEnt-IRL

+The authors prove the first finite guarantees for IRL with nonlinear reward functions

+The empirical results are promising and improve upon SOA

-Some of the claims of novelty seem to be standard practice and are used in prior work.

-It is unclear why an ML approach to IRL should perform better than other approaches. The authors have nice theory, but in practice the implemented algorithm seems very similar to prior work.

-It is unclear how statistically significant the empirical results are with only 3 seeds and no confidence intervals.

---

> ### Author Response · Authors · 2022-08-02
> **Author's Response: Contributions and Clarifications**
>
> We thank the reviewer for the detailed review of the paper and the valuable feedback. Below, we address the reviewer's comments in a point-by-point manner.
>
> **Weakness 1**: Some of the claims of novelty seem to be standard practice and are used in prior work.
>
> **Our Response**: The novelty of this work resides in the theoretical finite-time performance guarantees for the proposed algorithm. This is achieved by considering the maximum likelihood IRL problem as a bi-level optimization problem, which allows us to use a two-timescale stochastic approximation to develop a sample-efficient algorithm. Additionally, our obtained algorithm also has superior performance compared with a number of SOTA algorithms for IRL.
>
> While [1] also considered a maximum likelihood IRL formulation, the algorithm used in that paper for computing the estimates has a nested loop structure. Hence, such an algorithm is ill-equipped to deal with the computational burden required by computing optimal policies with functional approximations for the value function in high-dimensional state space. We would also like to call the reviewer’s attention to the quality of the rewards’ estimates reported in the cited paper [1] (see page 3956)– the implemented algorithms in [1] produce vastly different reward estimates even for a “toy” Gridworld problem.
>
> This clearly indicates there are issues with the algorithms used for computing the parameter estimates. In contrast our proposed algorithm has strong theoretical guarantees. For example, with linear reward parameterization (as in the Gridworld environment) the proposed algorithm is guaranteed to identify the global optimal reward estimator in finite time; see our discussions after Theorem 1 and Theorem 2 for detailed statement of this claim. We are not aware of any other works on IRL with this kind of theoretical guarantee.
>
> [1] Jain, Vinamra, Prashant Doshi, and Bikramjit Banerjee. "Model-free IRL using maximum likelihood estimation." Proceedings of the AAAI Conference on Artificial Intelligence. Vol. 33. No. 01. 2019.
>
> **Weakness 2**: It is unclear why an ML approach to IRL should perform better than other approaches. The authors have nice theory, but in practice the implemented algorithm seems very similar to prior work.
>
> **Our Response**: Thanks for this good question. We will answer why ML-IRL is better than other approaches, by comparing three lines of works:
>
> * The proposed algorithm is unlike algorithms for MaxEnt-IRL. This is because MaxEnt_IRL algorithms have a nested loop structure that requires the solution of an MDP in the inner loop. In contrast, the proposed algorithm has a single loop structure. In Appendix B.3, we have added a new experiment, which compares the convergence speed between our proposed algorithm and MaxEnt-IRL. From this experiment, it is obvious that ML-IRL and MaxEnt-IRL could obtain similar reward estimators while ML-IRL is much more efficient.
>
> * Guided cost learning (GCL) is a single-loop algorithm for computing the maximum likelihood estimator in IRL. In GCL, samples from the current estimate of the state-action trajectory distribution are used to estimate the reward gradient. This is a form of importance sampling.  The computational burden of sampling paths with long planning horizons and high-dimensional state space is significant. Hence, implementations of GCL use ad-hoc approximation techniques that often induce instabilities. It must be emphasized that GCL has no theoretical guarantees for finite-time performance. In contrast, in the proposed algorithm ML-IRL, the inner loop is replaced with a policy update under the entropy-regularized MDP defined by the current reward parameter estimate. Moreover, ML-IRL samples trajectories from the updated policy for estimating the reward gradient.  Note that sampling from a Markovian policy is simpler than sampling from a trajectory distribution as in GCL. In addition, we could easily use state-of-the-art methods like soft Q-learning or soft Actor-Critic to implement the policy update. Finally, let us emphasize again that, our proposed algorithm has a finite-time performance guarantee, while to our knowledge, algorithms for GCL do not have comparable results.
>
> * When compared with imitation learning algorithms, the better performance exhibited by the proposed algorithm should not come as a surprise. Like MaxEnt-IRL methods, the proposed method aims to recover the structure of the MDP whose optimal policy best matches the data. The structure of such MDP consists of both the reward function and the associated optimal policy. In contrast, imitation learning only aims to find policies matching the expert’s. Hence, the proposed method is better equipped to deal with counterfactual scenarios such as one with a different environment as presented in the paper. In these scenarios, the policies learned by imitation learning methods have no factual basis to estimate the optimal courses of action.

---

> > ### Author Response · Authors · 2022-08-02
> > **Response Continued: Contributions and Clarifications**
> >
> > **Weakness 3**: It is unclear how statistically significant the empirical results are with only 3 seeds and no confidence intervals.
> >
> > **Our Response**: Thanks for this comment. Now we have added up to 6 random seeds. Due to the limited space, we present the numerical results with confidence intervals in Appendix (see Table 3).
> >
> > **Question 1**: The authors are missing related work that also seeks to make IRL more tractable.
> >
> > **Our Response**: Many thanks for bringing us these interesting papers. We include these related works in our current version (see Appendix A).
> >
> > **Question 2**: One of the stated contributions is to implement a single-loop approach. This doesn't seem like a contribution since most Adversarial IRL methods do the same thing. This idea is well studied, going back to GAIL and Guided cost learning, and probably earlier. I do not think the authors can claim this as novel. Furthermore, the two-timescale approach is common in GANs and GAIL-like algorithms. I don't think this can be claimed as a contribution.
> >
> > **Our Response**: As we have emphasized above, the major contribution in our work is that we develop the **first provable finite-time algorithm** for **IRL problems**, under nonlinear reward parameterization. Although the two-timescale / single-loop approaches have been adopted in other algorithms like GAN, GAIL and GCL, none of these algorithms can be directly extended to the IRL setting while still maintaining provable theoretical guarantees. Moreover, our proposed algorithm substantially outperforms GAIL-like algorithms and GCL in the practical performance.
> >
> > **Question 3**: Equation 3: The theta parameterizes r which informs argmax to get pi. This is confusing with line 119 where the policy is parameterized by theta. Policies are not usually parameterized by rewards.
> >
> > **Our Response**: Thanks for this good question. We have revised our paper in line 119 to clarify that the policy is not parameterized by $\theta$.
> >
> > Since the optimal policy of an entropy-regularized MDP is unique under each reward parameter $\theta$, we denote such optimal policy as $\pi_{\theta}$ when the reward function is parametrized by parameter $\theta$.
> >
> > **Question 4**: Line 155: is supposed to be an example of solving a hard optimization problem in the inner loop but it refers to IQ-Learn which was previously noted to be computationally efficient because it avoids this inner loop mdp solver.
> >
> > **Our Response**: Thanks for this good comment. We have revised our paper in line 155.
> >
> > We claim that algorithms for MaxEnt-IRL suffer from the computational burden in their nested loop computation. Moreover, recently proposed algorithms like IQ-Learn try to improve the computational efficiency by sacrificing the estimation accuracy of the recovered rewards.
> >
> > In contrast to these algorithms, the goal of our design is to find provably efficient algorithm which could avoid high-complexity operations and accurately recover the reward function.

---

> > > ### Author Response · Authors · 2022-08-02
> > > **Response Continued: Contributions and Clarifications**
> > >
> > > **Question 5**: Lemma 1: How is this different from the guided cost learning update. In guided cost learning the partition function is estimated with onpolicy rollouts so it appears quite similar.
> > >
> > > **Our Response**: Thanks for this good question. Our gradient expression is quite different from that of GCL.
> > >
> > > First, let us show the gradient expression w.r.t. the reward parameter $\theta$ in GCL:
> > > $$ \nabla  {L}_{\rm GCL}(\theta) = \mathbb{E}\_{\tau \sim \mathcal{D}} [  {\sum}\_{t \geq 0} \gamma^t \nabla\_{\theta} r(s_t, a_t; \theta) ]  - \mathbb{E}\_{\tau \sim p\_{\theta}} [ {\sum}\_{t \geq 0} \gamma^t \nabla\_{\theta} r(s_t, a_t; \theta) ] $$
> > >
> > > From the gradient expression above, GCL needs to sample the trajectory from the trajectory distribution $p_{\theta}$ which is proportional to the exponential of the cumulative reward. In contrast, in Lemma 1 our algorithm samples the trajectory from the optimal policy $\pi_{\theta}$.
> > >
> > > As we mentioned in our response to Weakness 2, GCL needs to consider importance sampling and the ad-hoc approximation techniques in estimating the trajectory distribution often induce instabilities. For GCL, the computational burden of sampling paths with long planning horizons and high-dimensional state space is also significant. As a comparison to GCL, sampling trajectories from a Markovian policy in our proposed algorithm is much simpler than sampling from a distribution of state-action state-action trajectories as in GCL. Moreover, since we model the optimal policy as the solution of an entropy-regularized MDP, we could leverage state-of-the-art methods like soft Q-learning or soft Actor-Critic to implement the policy update, which also makes the problem more tractable.
> > >
> > > Finally we would like to emphasize that GCL does not have any theoretical guarantee of performance. In contrast, the algorithm proposed in this paper is the **first provable algorithm with finite-time analysis** for **IRL under nonlinear reward parameterization**.
> > >
> > > **Question 6**: Line 188: Why only sample one demo and one policy rollout? Doesn't this have very high variance?
> > >
> > > **Our Response**: Thanks for this good question. We answer this question from two perspectives.
> > >
> > > * From the theoretical perspective, we analyze the most general case where a stochastic estimator of the reward gradient is generated at each update step. Therefore, even though we sample a small batch of trajectories to update the reward parameter, the same stochastic estimator can still be constructed (use a batch of trajectories rather than one trajectory). The only difference may be that the variance of the stochastic gradient estimator is dependent on the sample size, but in our analysis we treat it as a bounded constant whose order is $\mathcal{O}(1)$. Therefore, as long as the variance stays in the same order, our convergence results still apply.
> > >
> > > * From the empirical perspective, in our experiment we actually sampled a small batch of trajectories each time to reduce the variance. We have updated this detail in Appendix B.
> > >
> > > We believe this interesting question could motivate a future research direction of this work:
> > >
> > > How to leverage variance reduction techniques to construct a good gradient estimator of the reward parameter, so that the convergence rate and practical performance of the algorithm could be improved?
> > >
> > >
> > > **Question 7**: Allowing the algorithm to work with state only demonstrations is nice, but many other algorithms already do this. E.g. Brown, Daniel S., Wonjoon Goo, and Scott Niekum. "Better-than-demonstrator imitation learning via automatically-ranked demonstrations." Conference on robot learning. PMLR, 2020. Torabi, Faraz, Garrett Warnell, and Peter Stone. "Generative adversarial imitation from observation." arXiv preprint arXiv:1807.06158 (2018). As well as all of the classical IRL algorithms.
> > > It would be good to motivate more why this is a contribution.
> > >
> > > **Our Response**: The compatibility of our algorithm with the state-only / transfer learning setting is not our main contribution, but a desirable property that our proposed algorithm enjoys.
> > >
> > > Since the identifiability of state-only reward in MaxEnt-IRL has been theoretically analyzed in the literature [2,3] and ML-IRL is close to MaxEnt-IRL, we expect that ML-IRL could also work well under the state-only / transfer learning setting. Our experiment matches our expectation and shows that ML-IRL could also outperform the benchmarks in the state-only / transfer learning setting.
> > >
> > > [2] Kim, Kuno, et al. "Reward identification in inverse reinforcement learning." International Conference on Machine Learning. PMLR, 2021.
> > >
> > > [3] Cao, Haoyang, Samuel Cohen, and Lukasz Szpruch. "Identifiability in inverse reinforcement learning." Advances in Neural Information Processing Systems 34 (2021): 12362-12373.

---

> > > > ### Comment · Reviewer_fBA5 · 2022-08-08
> > > > **response**
> > > >
> > > > Thank you for your response. I am still not convinced that there is much difference between the GCL update and the proposed approach. In the GCL paper Section 4.2, they note that they can use a maximum entropy objective to sample from the required distribution. This seems very similar to the approach taken in the submission which also uses a maximum entropy policy optimization (3a).
> > > >
> > > > If the optimal policy in the proposed approach is simply maximizing the current best guess of the reward function + an entropy bonus, then this still appears to be the same as GCL.
> > > >
> > > > I agree that the theoretical contributions are novel. My main concern was that the paper, as written, seemed to imply many more contributions such as the the two-time scale optimization.

---

> > > > > ### Author Response · Authors · 2022-08-09
> > > > > **Clarifications to the Follow-Up Comments**
> > > > >
> > > > > Thank you for your feedback! We would like to do further clarifications to show the difference between GCL [1] and ML-IRL in their formulations and their policy optimization methods.
> > > > >
> > > > > * **Problem Formulations**
> > > > >
> > > > > >(GCL)   $ ~~~~ \max\_{\theta} ~\mathbb{E}\_{\tau \sim \mathcal{D}}[ \log(p\_{\theta}(\tau)) ] ~~(a1)$
> > > > > >$ ~~~~~~~~~~~~~~ \text{s.t. } ~ p\_{\theta} :=  \arg \max\_{p} ~ \mathbb{E}\_{\tau \sim p(\cdot)} \big[ \mathcal{H}(p(\cdot)) + R(\tau; \theta) \big]  ~~(a2)~~~~~~~~~~~~~~~~~~~~~~~~~~~~~~~~~~~~~~~~~~~~~~~~~~~~~~~~~~~~~~~~~~~~~~~~~~~~~~~~~~~~~~~~~~~~~~~~~~~~~~~~~~~~~~~~~~~~~~~~~~~~~~~~~~~~~~~~~~~~~~~~~~~~~~~~~~~~~~~~~~$
> > > > > >$~~~~~~~~~~~~~\text{where } R(\tau; \theta) := \sum\_{t=0}^{\infty} [r(s_t, a_t; \theta)]$ and $\mathcal{H}(p(\cdot)) := -\sum_{\tau} p(\tau) \log p(\tau)$
> > > > >
> > > > > >(ML-IRL)   $ ~ \max\_{\theta} ~ \mathbb{E}\_{\tau \sim \mathcal{D}}[ \sum\_{t=0}^{\infty} \gamma^t \log(\pi\_{\theta}(a_t|s_t) ) ] ~~(b1)$
> > > > >  >$ ~~~~~~~~~~~~~~~ \text{s.t. } \pi\_{\theta} := \arg \max\_{\pi} ~ \mathbb{E}\_{\tau \sim \pi} \Big[ \sum\_{t=0}^{\infty} \gamma^t \big(r(s_t, a_t; \theta) + \mathcal{H}(\pi(\cdot | s_t)) \big) \Big] ~~(b2)$
> > > > >
> > > > > From these formulations, we believe there is significant difference between GCL and ML-IRL in the models of agent’s behavior.
> > > > >
> > > > > GCL takes an “open-loop” approach as it directly models the trajectory distribution as a Boltzmann distribution where $p\_{\theta}(\tau) \propto \exp\big( \sum_{t=0}^{\infty} r(s_t, a_t; \theta) \big)$. GCL leverages the guided policy search [2] in order to solve the entropy-regularized trajectory optimization problem (see equation (a2) above) and obtain the trajectory distribution $p\_{\theta}$.
> > > > >
> > > > > In contrast, ML-IRL takes a “closed-loop” approach: it models the agent’s behavior with a Markovian “feed-back” policy $\pi\_{\theta}(a|s) \propto \exp \big( Q^{\rm soft}_{r\_{\theta}, \pi\_{\theta}}(s,a) \big)$. ML-IRL leverages the state-of-the-art RL algorithms (soft Q-learning [3], soft Actor-Critic [4]) to solve the maximum entropy reinforcement learning problem in equation (b2) to obtain the policy $\pi\_{\theta}$.
> > > > >
> > > > > Differences in modeling agent’s behavior have important computational implications.
> > > > > When the planning horizon is large and the state space is high dimensional, modeling the agent’s behavior via a distribution of trajectories is not practical, likely leading to unstable training. To verify this point, we have conducted an additional set of experiments on CartPole where we set the horizon as 200. Please check the numerical results: https://imgur.com/dHBBdTf
> > > > >
> > > > >  Although this is a toy example, we can still observe that ML-IRL achieves faster and more stable convergence compared with GCL. (Note that there is no official implementation of the GCL which follows the descriptions in [1] to use guided policy search for policy optimization. Therefore, we use a public implementation of GCL in github: https://github.com/nishantkr18/guided-cost-learning)
> > > > >
> > > > > Another fact that demonstrates the superiority of a model based upon a “closed-loop” policy (as opposed to a model based upon a distribution of “open-loop” trajectories) is that state-of-the-art algorithms in RL [3,4] have used such model (entropy-regularized reinforcement learning problem (b2)). This also explains why most algorithms in imitation learning and inverse reinforcement learning [5,6,7] use a “closed-loop”  policy model and not a model based upon “open-loop” trajectories.
> > > > >
> > > > > In the end, we would like to quote a statement from the paper of GAIL [5] which has pointed out the weakness in the practical implementation of GCL: “Guided cost learning [1], for instance, builds upon guided policy search [2] and inherits its sample efficiency, but also inherits its requirement that the model is well-approximated by iteratively fitted time-varying linear dynamics. ”
> > > > >
> > > > > [1] Finn, Chelsea, Sergey Levine, and Pieter Abbeel. "Guided cost learning: Deep inverse optimal control via policy optimization." International conference on machine learning. PMLR, 2016.
> > > > >
> > > > > [2] Levine, Sergey, and Pieter Abbeel. "Learning neural network policies with guided policy search under unknown dynamics." Advances in neural information processing systems 27 (2014).
> > > > >
> > > > > [3] Haarnoja, Tuomas, et al. "Reinforcement learning with deep energy-based policies." International conference on machine learning. PMLR, 2017.
> > > > >
> > > > > [4] Haarnoja, Tuomas, et al. "Soft actor-critic algorithms and applications." arXiv preprint arXiv:1812.05905 (2018).
> > > > >
> > > > > [5] Ho, Jonathan, and Stefano Ermon. "Generative adversarial imitation learning." Advances in neural information processing systems 29 (2016).
> > > > >
> > > > > [6] Fu, Justin, Katie Luo, and Sergey Levine. "Learning robust rewards with adversarial inverse reinforcement learning." arXiv preprint arXiv:1710.11248 (2017).
> > > > >
> > > > > [7] Garg, Divyansh, et al. "IQ-Learn: Inverse soft-Q Learning for Imitation." Advances in Neural Information Processing Systems 34 (2021): 4028-4039.

---

> > > > > > ### Comment · Reviewer_fBA5 · 2022-08-09
> > > > > > **response**
> > > > > >
> > > > > > Thank you for your clarification and detailed explanation and for running an additional experiment. I agree that the open-loop vs closed-loop distinction is important.

---

> ### Author Response · Authors · 2022-08-07
> **Looking Forward to Post-Rebuttal Feedback**
>
> Dear reviewer fBA5,
>
> Thank you very much for taking the time to review our paper! We cherish your comments and evaluations very much! In our posted responses, we have made a point-to-point response to alleviate your concerns. If you have any further concerns on our response, we are happy to address them.
>
> Best,
>
> Authors

---

### Meta-Review · Area_Chair_Sbzj · 2022-08-27

**Recommendation:** Accept
**Confidence:** Less certain

**Metareview:**

This paper presents a new single loop IRL algorithm that avoids the typical policy/reward optimization loop in IRL algorithms, without sacrificing the accuracy of the learned reward function. This is achieved through the use of stochastic gradients of the likelihood function. The proposed algorithm is proved to converge to a stationary solution with a finite-time guarantee. Experiments on some problems in MuJoCo show that the proposed algorithm can outperform existing solutions. The reviewers all agree that the paper is well-written, the algorithm is sufficiently new, and the experiments are compelling. There are some concerns that experimental evaluation is limited to MuJoCo locomotion tasks which can sometimes be too simple to draw strong conclusions.

**Award:**

No

---

### Decision · Program_Chairs · 2022-09-14

Accept